# Molecular hydrogen in minerals as a clue to interpret ∂D variations in the mantle

B. N. Moine [1]✉, N. Bolfan-Casanova[2], I. B. Radu[1,3], D. A. Ionov[4], G. Costin [5], A. V. Korsakov[6], A. V. Golovin[6,7], O. B. Oleinikov[8], E. Deloule[9] & J. Y. Cottin[1]

Trace amounts of water dissolved in minerals affect density, viscosity and melting behaviour of the Earth's mantle and play an important role in global tectonics, magmatism and volatile cycle. Water concentrations and the ratios of hydrogen isotopes in the mantle give insight into these processes, as well as into the origin of terrestrial water. Here we show the presence of molecular $H_2$ in minerals (omphacites) from eclogites from the Kaapvaal and Siberian cratons. These omphacites contain both high amounts of $H_2$ (70 to 460 wt. ppm) and OH. Furthermore, their ∂D values increase with dehydration, suggesting a positive H isotope fractionation factor between minerals and $H_2$–bearing fluid, contrary to what is expected in case of isotopic exchange between minerals and $H_2O$-fluids. The possibility of incorporation of large quantities of H as $H_2$ in nominally anhydrous minerals implies that the storage capacity of H in the mantle may have been underestimated, and sheds new light on H isotope variations in mantle magmas and minerals.

[1] Université de Lyon, UJM-Saint-Etienne, UCA, IRD, CNRS, Laboratoire Magmas et Volcans, UMR6524, Saint-Etienne, France. [2] Laboratoire Magmas et Volcans, Université Clermont-Auvergne, CNRS UMR 6524, Clermont-Ferrand IRD R 163, France. [3] Department of Geological Sciences, University of Cape Town, Rondebosch, Cape Town 7701, South Africa. [4] Géosciences Montpellier, Université de Montpellier, Montpellier 34095, France. [5] Department of Earth, Environmental and Planetary Sciences, Rice University, Houston, TX 77005, USA. [6] Sobolev Institute of Geology and Mineralogy, Siberian Branch Russian Academy of Sciences (SB RAS), Koptyuga 3, Novosibirsk 630090, Russia. [7] Novosibirsk State University, Pirogova 2, Novosibirsk 630090, Russia. [8] Diamond and Precious Metal Geology Institute, SB RAS, Yakutsk 677007, Russia. [9] CRPG, UMR7358, CNRS, Université de Lorraine, Vandoeuvre-lès-Nancy, France. ✉email: bertrand.moine@univ-st-etienne.fr

Hydrogen (or water in its oxidised form) plays a key role in the evolution, dynamics and habitability of the Earth. Even in minor amounts, it decreases the mechanical strength and melting temperatures of rocks and minerals, properties that govern volcanism and mantle convection. Hydrogen was incorporated into the Earth's interior during its accretion[1–3] and then evolved through degassing by volcanism and recycling by subduction. It is an ubiquitous trace component of nominally anhydrous minerals (NAMs) in the upper mantle, estimated to amount, as water, to 0.5–1 times the mass of the oceans[4,5], with cosmochemical arguments leading to an estimate of up to seven oceanic masses in the initial Bulk Silicate Earth[6]. So far, hydrogen was thought to exist in the mantle in the form of hydroxyl (OH) with storage capacity depending on depth[7,8]. An equivalent to a few hundred ppm by weight of $H_2O$ was measured in peridotite xenoliths (mantle fragments brought up by volcanic eruptions)[9,10] and up to 1.2 wt% $H_2O$ in a ringwoodite inclusion in an ultradeep diamond from the transition zone[11]. On the other hand, the studies of eclogite xenoliths from Slave, West African and Zimbabwe cratons indicate that oxygen fugacity, $fO_2$, ranges from $\Delta\log fO_2$ −2 to −4.5[12] relative to the Fayalite–Magnetite–Quartz buffer[13,14]. At such fugacity, a substantial amount of H should be present in a reduced form[15]. Reducing conditions were demonstrated experimentally to greatly decrease the solubility of OH in olivine[16] and it was recently discovered that $H_2$ could also be dissolved in NAMs, while it had remained undetectable due to its low infra-red extinction coefficient[17].

Here we report H concentration, speciation and isotope ratios for omphacite (sodic clinopyroxene) from 12 eclogite xenoliths from the Kaapvaal (Roberts Victor Mine) and Siberian (Obnazhennaya kimberlite) cratons. These eclogites are bi-mineralic (omphacite, garnet) and corundum-bearing rocks (Fig. 1) with estimated equilibrium conditions of 2.1−3.4 GPa, 805−1110 °C, and 2.8–4.1 GPa, 923−1140 °C, respectively (Supplementary Table 1). They are believed to represent subducted oceanic crust preserved in the cratonic root for >2 billion years and unaffected by host kimberlite or older melts[18]. The samples were analysed for their hydrogen abundances (expressed as water) and isotopic composition by Thermal Conversion-Elemental Analyser coupled with continuous flow mass spectrometer (TC/EA-IRMS)[19,20]. Hydrogen was also measured in the omphacite by SIMS (Secondary Ion Mass Spectrometry) and FTIR (Fourier Transform Infra-red Spectroscopy). The hydrogen abundances obtained by TC/EA-IRMS and SIMS (that detect all forms of hydrogen) are consistently higher than those obtained by FTIR (that only detects OH and $H_2O$ efficiently), see Table 1. We propose that the

**Table 1 Water content and hydrogen isotope composition of omphacites.**

| Sample | n | $\delta D‰$ vSMOW | ±‰ | $H_2O$ (ppm) TC/EA-MS | 1 SD | n | $H_2O_{tot}$ (ppm) FTIR 3000–3800 cm$^{-1}$ | ±30% | $H_2O$ (ppm) SIMS | 1SD | Abs norm 5200 cm$^{-1}$ | $H_2O_{mol}$ (ppm) FTIR 5200 cm$^{-1}$ | Abs Int norm 4000–4300 cm$^{-1}$ | $H_2$ (ppm) | 1 SD |
|---|---|---|---|---|---|---|---|---|---|---|---|---|---|---|---|
| Obn108 | 12 | −107 | 11 | 2750 | 100 | 7 | 1177 | 353 | 4850 | 340 | 0.0536 | 178 | 4.62 | 175 | 41 |
| Obn110 | 12 | −120 | 11 | 4850 | 260 | 6 | 1249 | 375 | | | 0.0417 | 138 | 6.02 | 400 | 51 |
| Obn11 | 9 | −116 | 4 | 4650 | 540 | 9 | 1413 | 424 | | | 0.0425 | 141 | 9.42 | 360 | 76 |
| Obn112 | 9 | −126 | 5 | 5065 | 340 | 12 | 929 | 279 | | | 0.0520 | 172 | 14.1 | 460 | 49 |
| RV179 | 1 | −92 | | 720 | 120 | 9 | 120 | 36 | | | | | | 67 | 4 |
| RV203 | 3 | −92 | 2 | 1315 | 130 | | | | | | | | | | |
| RV233 | 5 | −99 | 5 | 1770 | 200 | 9 | 354 | 106 | | | | | | 157 | 25 |
| RV360 | 5 | −107 | 2 | 2400 | 300 | 9 | 123 | 37 | | | | | | 253 | 34 |
| RV377 | 4 | −93 | 3 | 1040 | 10 | | | | | | | | | | |
| RV469 | 3 | −99 | 2 | 1467 | 65 | | | | | | | | | | |
| RV488 | 3 | −91 | 2 | 1230 | 280 | 9 | 173 | 52 | | | | | | 117 | 32 |
| RV513 | 4 | −89 | 3 | 1256 | 256 | 9 | 255 | 77 | | | | | | 111 | 30 |

Water content and H isotopic composition were measured using a Thermal conversion/Elemental Analyser coupled with Isotope-Ratio Mass spectrometer (TC/EA-IRMS), Fourier Transform Infra Red (FTIR) and Secondary Ion Mass Spectrometry (SIMS) analyses. FTIR concentrations were calculated using previously published absorption coefficient[34]. Molecular water content was estimated based on the specific IR absorption peak (5200 cm$^{-1}$), accounting for both OH and potential $H_2O_{molecular}$ content (TC/EA-IRMS) and FTIR integrated intensity in the range of 3000–3800 cm$^{-1}$, and the absorption coefficient of[27]. Molecular $H_2$ was calculated by difference between total $H_2O$ content (TC/EA-IRMS) and FTIR integrated intensity in the range of 3000–3800 cm$^{-1}$. Abs. norm absorbance corresponding to 5200 cm$^{-1}$ peak height, 1 SD standard deviation, n number of analyses

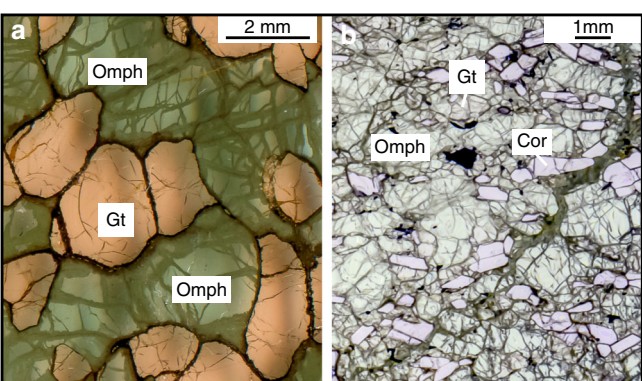

**Fig. 1 Thin section images of eclogite samples. a** bimineralic eclogite from Roberts Victor Mine (South Africa) and **b** corundum-bearing eclogite from Obnazhennaya kimberlite (Siberia).

"missing" hydrogen undetected by FTIR is stored in minerals as molecular $H_2$, which implies that current estimates for total hydrogen in the mantle may be too low. Our results also show that $H_2$ concentrations are correlated with H isotope compositions, providing new insights into H-isotope systematics in mantle rocks and minerals.

## Results and discussion

**H contents**. The water concentrations (Table 1) in the omphacite measured by TC/EA-IRMS range from 720 to 5065 wt ppm, consistent with data for orogenic (crustal) eclogites (1200−6000 ppm[21−23]), and are generally much higher than the maximum of 600 ppm $H_2O$ reported previously for mantle pyroxenes[10,24,25]. Thus, eclogites could represent a significant reservoir of water in the cratonic lithosphere, despite their relatively low average abundance (2%)[22,23], as well as generally in the convecting mantle[26]. FTIR spectra in the OH region (2800−3800 cm$^{-1}$; Fig. 2) systematically indicate much lower water contents (100−1500 ppm $H_2O$) than those obtained by TC/EA-IRMS or SIMS (Table 1). The most important difference between the two methods is that TC/EA-IRMS records the totality of H atoms whereas the FTIR value only relates to structurally bound OH and molecular water[27]. This suggests that some hydrogen is stored in the omphacite not only as OH but in a different form. Discrepancies between bulk and spectroscopic methods have already been observed for omphacites in orogenic eclogites and magmatic clinopyroxene[21,28] and tentatively explained by the presence of nano-bubbles[29,30] of molecular water in the minerals. Although previous work did not identify molecular water spectroscopically, except maybe through the band at 3400 cm$^{-1}$ [31], we observed it in some of the analysed grains, albeit with a very noisy signal. Its contribution is very low, 140−290 ppm, within the error of TC/EA-IRMS and FTIR measurements (Table 1), estimated based on the 5200 cm$^{-1}$ specific band[32] using published absorbance coefficients[27]. Molecular water, therefore, cannot account for the measured H excess. It thus looks like most of the total water in minerals measured in the 2800−3800 cm$^{-1}$ range is present in the form of hydroxyl (OH). The SIMS $H_2O$ measurements for sample Obn110 are consistent with TC/EA-IRMS data (4850 ppm, Table 1). Since SIMS is a microbeam technique (that measures all H atoms independently of their speciation), the fact that the water content it provides agrees with the bulk water measurements by TC-EA/IRMS indicates that the amount of H in occasional micro-inclusions and/or fractures is not significant.

**H speciation**. We also detected a very small and pleochroic peak in the infra-red spectra near 4100 cm$^{-1}$ for the most water-rich omphacites from corundum-bearing Obnazhennaya eclogites that contain 2750−5065 wt ppm water based on the TC/EA-IRMS method (Fig. 2). This peak was previously proposed to correspond to molecular $H_2$[17]. Molecular $H_2$ does not normally respond to infra-red excitation due to its symmetry. However, if $H_2$ is dissolved in an ionic environment, the weak forces cause the appearance of a dipole interacting with infra-red radiation[33]. The centroid of this peak is reduced by ~50 cm$^{-1}$ compared with that of molecular $H_2$ vapour determined by Raman spectroscopy[17]. We calculated the amount of molecular $H_2$ (70−460 wt ppm; Table 1) by difference between the FTIR and TC/EA IRMS data, and found H/OH molar ratios > 3.

It is further possible to estimate the IR absorption coefficient for $H_2$ from the FTIR absorbance areas and $H_2$ contents determined above using the Beer-Lambert law (see "Methods" and Supplementary Fig. 1). The fact that the concentration of molecular $H_2$ calculated as the difference between bulk water obtained from TC-EA-IRMS and OH obtained from FTIR

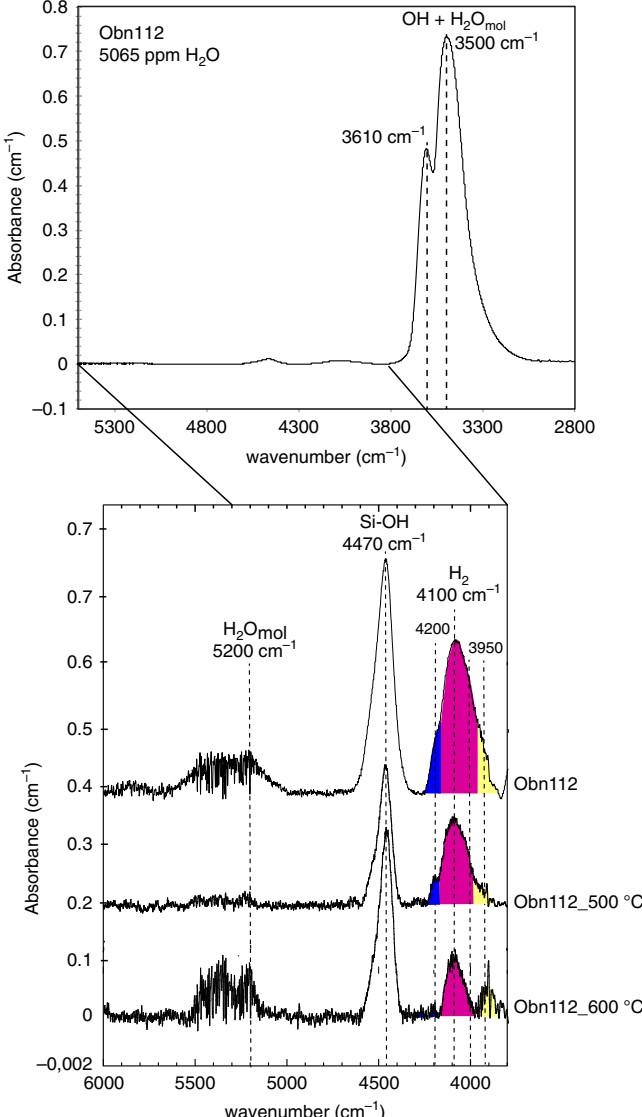

**Fig. 2 Unpolarised FTIR spectra of omphacite (Obn112 sample).** In the Fourier Transform Infra Red (FTIR) spectra, the 2800–5500 cm$^{-1}$ range show main peaks in the 3000–3800 cm$^{-1}$ range corresponding to OH and molecular $H_2O$ (stretching). Secondary peaks in the 3800–6000 cm$^{-1}$ interval correspond to molecular $H_2$ (4100 cm$^{-1}$), Si-OH (4470 cm$^{-1}$), and molecular water (5200 cm$^{-1}$). A comparison between unheated and heated samples (500 and 600 °C) shows decreasing intensity of the peak at 4100 cm$^{-1}$ with increasing temperature. The widening of the peak with increasing $H_2$ content is due to overlapping with two smaller peaks at 4200 and 3950 cm$^{-1}$. Note that the low intensity of the band at 5200 cm$^{-1}$ related to molecular water indicates negligible fluid inclusion contributions.

correlates positively with the integrated absorbance indicates that the assignment of the 4100 cm$^{-1}$ peak to $H_2$ is correct. Furthermore, the average linear absorption coefficient of $H_2$ that we calculate (Table 1) is 1500 times lower than for OH (0.13 ± 0.3 l mol($H_2$)$^{-1}$ cm$^{-1}$ or ~44 ± 10 l mol($H_2$)$^{-1}$ cm$^{-2}$ for $H_2$ vs. 65,000 l mol$^{-1}$ cm$^{-2}$ for OH[34]). This is much lower than those previously proposed (46.4 l mol($H_2$)$^{-1}$ cm$^{-1}$ or 1650 l mol($H_2$)$^{-1}$ cm$^{-2}$)[17] for NAMs (orthopyroxene) annealed under reducing conditions, but in agreement with previous measurements on silica (0.26 l mol($H_2$)$^{-1}$ cm$^{-1}$)[33]. This further highlights the very low infra-red activity of $H_2$ and therefore the very low detectability of molecular $H_2$ by spectroscopic methods. This

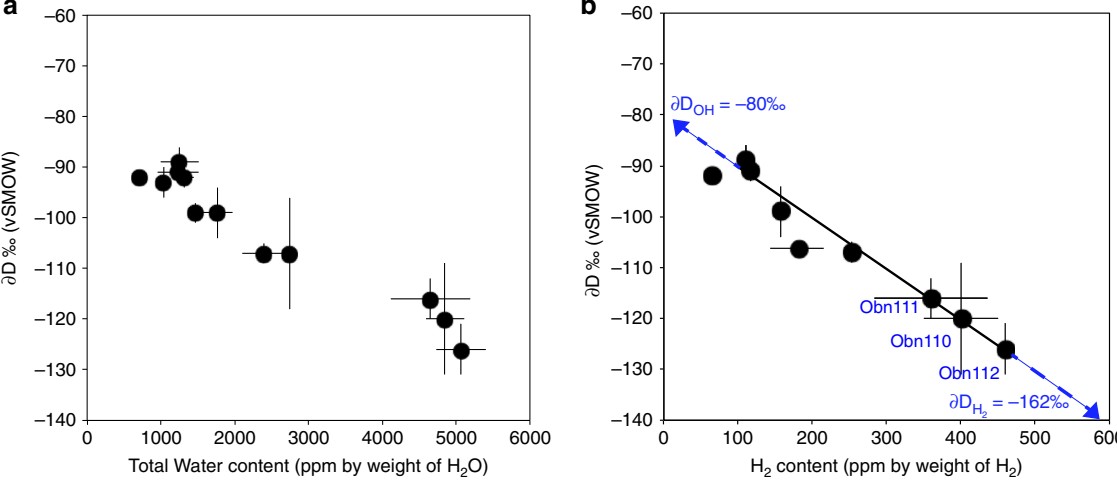

**Fig. 3 Hydrogen isotope composition of omphacites versus H content. a** $\partial D$ (relative to Vienna Standard Mean Ocean Water – V-SMOW) versus total water content determined by Thermal conversion/Elemental Analyser coupled with Isotope-Ratio Mass spectrometer (TC/EA-IRMS, and **b** $\partial D$ (relative to V-SMOW) versus calculated $H_2$ content (wt ppm). Error bars correspond to 1 SD.

strongly suggests that the amounts of H determined in previously studied mantle xenoliths have been greatly underestimated.

**H isotopic composition**. We examine the effects of hydrogen speciation in minerals on H isotope fractionation in the deep water cycle. The $\partial D$ values in the omphacites from this study correlate well with H concentrations expressed either as $H_2O$ or as $H_2$ (Fig. 3a, b). The well-defined trend indicates an increase of the $\partial D$ values in the omphacites with decreasing water content. To confirm the speciation of H, Obnazhennaya samples were heated under vacuum ($10^{-3}$ mbar) at 400 °C, with sample Obn112 incrementally heated at 250, 400, 500, and 600 °C. After each heating step, that lasted 20 min, the samples were re-analysed with FTIR and TC-EA/IRMS. These stepwise experiments show that (1) the bulk water content and the isotopic composition of the samples are little affected by heating up to 400 °C (see Supplementary Table 2). Only two samples (Obn110 and Obn108) suffered up to 15% water loss during heating up to 400 °C. This means that if any inclusions were present, the amount of water stored in them must be negligible or within 15% because otherwise, like in fluid inclusion studies, or for garnets containing water-inclusions, the speciation and concentration of bulk water would be affected[31,35]. If we consider only the samples annealed at the highest temperature and consider them as the most free of the contribution from inclusions then we get an absorptivity coefficient of $30 \, l \, mol(H_2)^{-1} \, cm^{-2}$, instead of the average of $44 \, l \, mol(H_2)^{-1} \, cm^{-2}$ determined on the samples before heating at high temperature. (2) Incremental heating of sample Obn112 up to 600 °C yields an increase in the $\partial D$ values of residual H along with a decrease in OH and $H_2$ concentrations in omphacite (Fig. 4), in agreement with the general trend described/recorded by the samples. One way to explain this negative correlation between the concentration of total water and isotopic composition is by loss during heating of a component with more negative $\partial D$ values, such as $H_2$ or a mixture of $H_2$ and OH[36]. Indeed, the global partitioning of $H_2$ and OH between mineral and fluid would enrich the fluid in $^1H$ and the residual solid in $^2D$[36]. Previous reports of such negative correlation on eclogites from Dabie Sulu have been interpreted by the loss of isotopically light molecular water due to kinetic fractionation of H-D during dehydration in the course of exhumation[37]. Given that the ascent of kimberlites is very fast, we propose instead that the H-D isotopic fractionation is controlled by the presence of $H_2$.

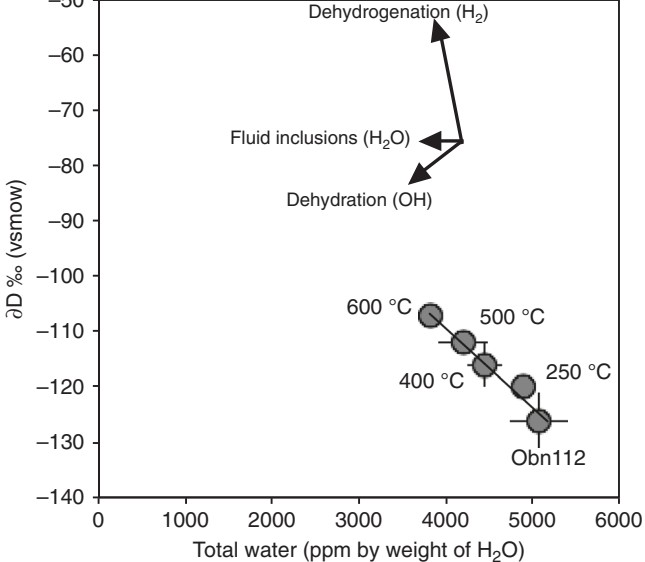

**Fig. 4 $\partial D$ versus $H_2O$ for heated and unheated omphacite Obn112.** Total water content (determined by Thermal conversion/Elemental Analyser coupled with Isotope-Ratio Mass Spectrometer - TC/EA-IRMS) and $\partial D$ (relative to Vienna Standard Mean Ocean Water – V-SMOW) show a robust linear correlation. The loss of H with increasing temperature implies higher $\partial D$ values supporting the loss of a component with much more negative $\partial D$ (thus $H_2$ rather than OH or $H_2O$), as also observed in the infrared spectra where the integrated absorbance of the $H_2$ band decreases significantly with increasing temperature.

Structural $H_2$ is indeed observed in omphacite, and the linear relationship between the absorbance of the 4100 cm$^{-1}$ band and the calculated $H_2$ content indicates that its quantification is robust (see Supplementary Fig. 1). Molecular water present in nano-inclusions seems to be negligible in these samples. However, if present in large quantities, nano-inclusions could also be filled with $H_2$ given that the conditions of equilibration of the present eclogites are very close to the conditions where $H_2$ and $H_2O$ are miscible within the mantle[38].

As shown in Fig. 2, the absorbance of the 4100 cm$^{-1}$ band decreases with increasing temperature. This decrease is decoupled

from that of the band at ~4500 cm$^{-1}$ (assigned unambiguously to a Si-OH vibration) implying that these two bands cannot be attributed to the same species, i.e. the 4100 cm$^{-1}$ band is not the result of some combination of OH mode (see Supplementary Figs. 2, 3). Indeed, if these two bands were due to the same species the ratio of their intensities would stay constant, which is not the case: it varies depending on temperature (see Supplementary Fig. 3). While the integrated absorbance of the band at 4500 cm$^{-1}$ decreases, between 100 and 400 °C, that of the 4100 cm$^{-1}$ band increases (Supplementary Fig. 2). This translates into H$_2$ being produced while OH is being consumed. Thus, we interpret the contrasting behaviour shown in Supplementary Figs. 2, 3 as a transition from OH to H$_2$ during the heating stage at low temperatures following the oxidation-dehydrogenation shown in reaction 1, similar to what is inferred for natural or experimental samples[16,39,40]:

$$H_2O + 2\,FeO <=> H_2 + Fe_2O_3 \qquad (1)$$

Such behaviour also indicates that the kinetics of H$_2$ and H$^+$ diffusion are close in this temperature range and cross over at higher temperatures in agreement with previous measurements[41–44] (Fig. 5).

Reaction 1 is probably responsible for the stabilisation of H$_2$ in NAMs linked to a change in iron valence in ferro-magnesian silicates, via reduction of water by ferrous iron. Ferric iron solubility increases in clinopyroxene and garnet with increasing pressure[15]. Thus, we can expect that eclogitic clinopyroxenes containing 7000−16000 wt ppm Fe, with Fe$^{3+}$/ΣFe estimated at 20−30% due to a high jadeite component [Na$^+$(Al$^{3+}$Fe$^{3+}$) Si$^{4+}_2$O$^{2-}_6$][45], can easily incorporate the H$_2$ concentrations measured in this study via reaction 1. Such a reaction is common for dehydration metamorphism in subduction zones[39]. In the absence of available oxygen, another reaction producing H$_2$ has

been proposed for the formation of diamond in the cratonic mantle:

$$CH_4 = C^{diamond} + 2\,H_2. \qquad (2)$$

However, the preservation within the mantle of high amounts of H (>2000 ppm), considered highly mobile, at high temperatures and for a long time, needs to be explained. Diffusivity of H$_2$ in minerals is currently unknown but existing data for silicate glasses indicate that H$_2$ diffusion is not very fast, no faster than for H$^+$ at mantle conditions. For example, the diffusivity of H$_2$ in silica glass[43,44] is $9.3 \times 10^{-16}$ m$^2$ s$^{-1}$ at 23 °C and $2.4 \times 10^{-12}$ m$^2$ s$^{-1}$ at 250 °C. Using the activation energy of 44 kJ mol$^{-1}$ provided in these studies we calculate a diffusivity of H$_2$ of $1.5 \times 10^{-10}$ m$^2$ s$^{-1}$ at 600 °C or $1.0 \times 10^{-9}$ m$^2$ s$^{-1}$ at 1000 °C (Fig. 5). Such values are very similar to those determined for the rate of the oxidation reaction[46] as well as for OH diffusion in olivine[41] or diopside[42]. The validity of these estimates ultimately hinges on the knowledge of diffusion mechanisms for molecular H$_2$ in minerals (vacancy vs. interstitial diffusion, polaron or Franck-Turbull mechanisms), which are currently unknown. Considering that H$_2$ diffusivity experiments in silicate glass have shown a strong dependence (three orders of magnitude)[46] (Fig. 5) on H$_2$ partial pressure, the above diffusivity (hence H loss) estimates are likely to be exaggerated. Such diffusivity also implies that equilibrium is geologically fast at the mineral grain scale at moderate to high temperatures, yet high H concentrations could be maintained on the scale of oceanic crust fragments hundreds of metres or kilometres in size. If such eclogitic blocks are preserved for several Ga in the mantle, they will develop hydrogen zoning in terms of abundance and isotope ratios, reflecting their progressive re-equilibration with the ambient mantle.

The surface water cycle fractionates hydrogen isotopes, creating a wide range of isotopically distinct reservoirs, such as Greenland ice caps standard precipitation [∂D = −190‰], seawater [VSMOW ∂D ~ 0‰] and rainwater [∂D = 0−130‰]). The deep water cycle may fractionate hydrogen isotopes as well, but the processes involved are different. Upper mantle (MORB) magmas typically have uniform ∂D values of −60 ± 5‰[47], whereas oceanic island magmas, thought to come from the lower mantle, may have much lower ∂D down to −218‰[48]. The ∂D values for omphacites in this study range from −89 to −126‰; they are similar to those reported for orogenic eclogites from the ultra-high pressure (UHP) Sulu terrane (−82 to −128‰[18]), and lower than those for MORB-like sources (−60‰ ± 5)[47]. Both in our study and previous reports of analyses of UHP rocks by TC/ EA-IRMS, the isotopic composition of H is observed to increase with decreasing bulk H of omphacite. This contradicts previous inferences that ∂D decreases during subduction, hence with dehydration, typically from modern oceanic crustal segments with ∂D of −35 ± 15‰[49–51], to orogenic or cratonic eclogites (remnants of ancient subducted oceanic crust), with ∂D of −82 to −128‰[18]. Experimentally determined mineral-water H isotope fractionation factors are generally negative[52,53] implying that the hydrogen remaining in the slab becomes increasingly depleted in deuterium which preferentially partitions into expelled fluids. Subduction-related dehydration thus causes a decrease in ∂D in slab materials with depth[54], as modelled in Fig. 6 (blue trend). The overall mineral-fluid H isotope fractionation factor is difficult to estimate because it varies greatly depending on temperature, pressure and mineral species during subduction-related slab metamorphism. Nevertheless, known H-D isotope fractionation between water and common hydrous minerals (serpentine, amphibole, chlorite, epidote, zoisite, brucite, and clays) range from −10 and −77‰ in the 100−800 °C temperature range, with decreasing fractionation at higher temperatures. However, is dehydration the actual process taking place in the present

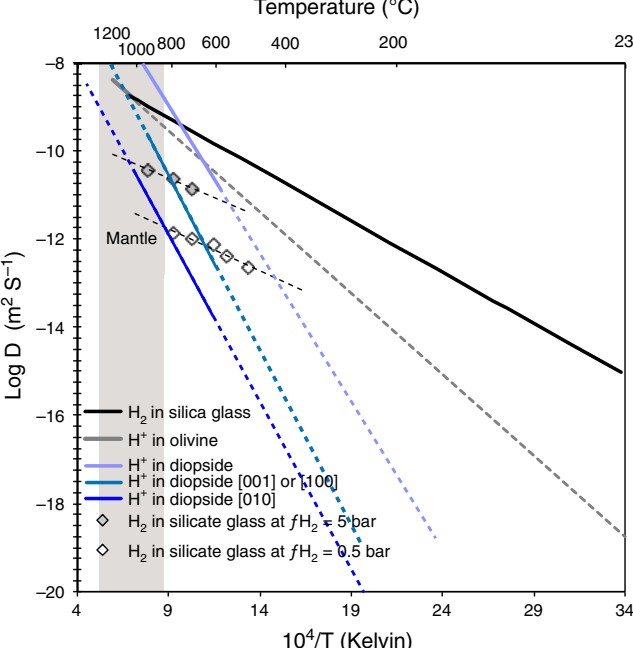

**Fig. 5 Experimental diffusivity of H in silicate materials.** Diffusivity of hydrogen in silica glass[43,44], effective diffusivity in olivine[41], diffusivity in diopside as a function of crystallographic orientation[42], and the effect of H$_2$ partial pressure on diffusivity in glass[46]. The grey field corresponds to the temperature range of the lithospheric mantle (dashed lines correspond to low temperature extrapolation of experimental data).

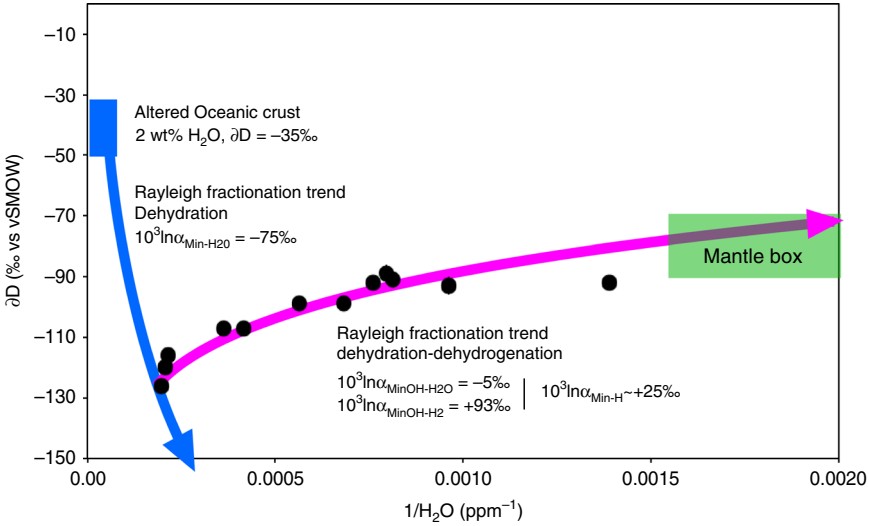

**Fig. 6 Model for evolution of H content and $\partial D$ in mantle omphacites.** Hydrogen isotope composition ($\partial D$ relative to Vienna Standard Mean Ocean Water—V-SMOW) versus $1/H_2O$ (water content determined by Thermal Conversion /Elemental Analyser coupled with Isotope Ratio Mass Spectrometer —TC/EA-IRMS). The blue trend models the compositional evolution of subducting oceanic crust dominated by dehydration. The magenta trend models the compositional evolution of oceanic material, during or post-subduction, dominated by dehydrogenation and dehydration. The model is based on Rayleigh fractionation. The blue box corresponds to the typical isotopic composition of oceanic crust altered by seawater[55,56] and the green box to the "normal" mantle values[47].

samples? In this work, we propose that dehydrogenation plays an important role in the isotopic compositions observed (see Fig. 6, green trend).

Here we model the change in the $D/H$ ratio of altered oceanic crust (starting at 2 wt% $H_2O$, $\partial D = -35$‰[55,56]) during subduction as it dehydrates. By using a Rayleigh fractionation process and a calculated fractionation factor[54] ($\alpha_{Mineral-H2O} = 0.9277$), it is possible to account for the highest water abundances and the highly negative $\partial D$ values, observed here for the cratonic omphacites ([$H_2O$] = 5065 ppm, $\partial D = -126$‰) (Table 1). However, such a low fractionation factor cannot explain why the $\partial D$ values increase upon dehydration from 5000 to 700 ppm water (Fig. 3a). The enrichment of a mineral in $^2D$ concomitant with dehydration can only be explained if the mineral-fluid fractionation factor is positive, which is the case for H partitioning between minerals and $H_2$[36]. In such a case, the de-volatilisation is accompanied by the release of a $H_2$ fluid enriched in $^1H$ instead of a $H_2O$ fluid enriched in $^2D$, which leads to less negative $\partial D$ values of residual hydrogen in the mineral. Since the isotope fractionation factor for molecular $H_2$ is positive and high[57], extraction of $H_2$ results in a positive mineral-($OH$-$H_2$) fractionation factor at high temperatures such that $\partial D$ values of the eclogite become less negative (Figs. 6, 7). Nonetheless, these isotopic compositions cannot be explained by $H_2$ release alone, which would induce a much faster isotopic evolution. The most likely scenario is a combination of dehydration and dehydrogenation processes (Figs. 6, 7) by diffusion during the very long residence time in the mantle of these eclogitic units as $H_2$ and $H^+$ have similar diffusivities at mantle temperature (Fig. 5).

The studied samples show a very good linear correlation between the calculated molecular $H_2$ content and $\partial D$ values of total H ($R^2$ of 0.93, $n = 12$, $p < 0.001$; Fig. 3b). This correlation between $H_2$ concentration and isotopic composition indicates that the $\partial D$ in omphacites reflects a mixture of two components, OH and molecular $H_2$, each having distinct isotopic compositions. From the proportions of molecular $H_2$ and the respective isotopic ratios of each sample, it is possible to determine the isotopic compositions of each end-member: $\partial D_{MineralOH} = -80$‰ and $\partial D_{MineralH2} =$

$-162$‰. We can then calculate the intra-mineral fractionation factor between OH and $H_2$ for the Obnazhennaya omphacites as follows: $\alpha_{MineralOH-MineralH2} = 1.098$ ($10^3 ln\alpha_{MineralOH-MineralH2} = +93$‰). Because hydrogen isotope fractionation between the structurally bound OH (MineralOH) and structurally bound $H_2$ (MineralH$_2$) is >20‰, the fractionation factor was calculated as follows: $\alpha_{A-B} = (1000 + \partial D_A)/(1000 + \partial D_B)$[53]. The intra-mineral fractionation between OH and $H_2$ calculated here is equivalent to the fractionation between molecular water and molecular hydrogen ($10^3 ln\alpha_{H2O-H2}$: ~+100‰) at very high temperature ($\geq 1100$ °C[57]). This suggests that high temperature isotopic equilibrium was reached and preserved in our samples. The diffusive loss of H during the transport of mantle xenoliths close to the surface by kimberlitic magma is low because the magma ascent is very fast[58].

Since molecular $H_2$ is most likely to be the dominant form of H in the reduced deep mantle (refs. [3,17]), it follows that isotopic fractionation of H in the mantle should be controlled by equilibria involving $H_2$-bearing minerals rather than $H_2O$- or OH-bearing minerals. This must be taken into account when interpreting the H isotopic distribution in the mantle and models involving deep mantle volatile loss[59]. Similar to findings of this study, clinopyroxene (augite) megacrysts from alkaline basalts at Nushan[28] yield different $H_2O$ contents by FTIR and TC/EA-IRMS (or manometrically), which correlate negatively with measured $\partial D$ (see Supplementary Table 1 and Supplementary Fig. 4). Assuming that the measured concentration difference is due to molecular $H_2$, the calculated $10^3 ln\alpha_{MineralOH-MineralH2}$ is estimated to be +111‰, which is realistic at magmatic temperatures. In addition, the presence of structurally bound $H_2$ could explain the large difference in $\partial D$ values of coexisting richterites ($-132$‰ vSMOW) and phlogopites ($-65$‰ vSMOW) in MARID xenolith suites from South African kimberlites, which was previously interpreted as a result of fractional crystallisation or re-equilibration during ascent[60]. Also, a recent discovery of highly negative $\partial D$ in deep magmas trapped in melt inclusions (e.g.[48]) could be due to different $H_2O_{tot}$-$H_2$ proportions and isotopic fractionation controlled by $fO_2$[61] or $H_2$ loss by diffusion, rather than a primary composition as previously believed. An

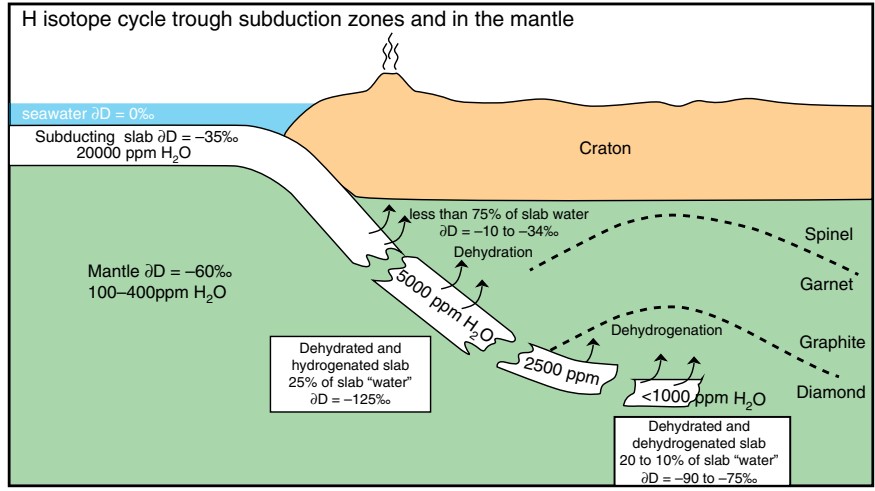

**Fig. 7 H recycling and isotope evolution in subduction zone.** The sketch shows how the recycled crust gets enriched in $^1H$ during subduction due to dehydration and subsequently gets enriched in D due to $H_2$-OH equilibration and diffusion within the upper mantle.

accurate determination of H speciation in mantle samples, allowing the quantification of hydrogen in molecular and hydroxyl forms, is therefore a prerequisite for any isotopic measurement and interpretation.

In this study we provide three main pieces of evidence for $H_2$ in minerals: (i) the discrepancy between hydrogen contents from mass spectrometry and FTIR, (ii) the presence of an absorption band in the infra-red spectra at $4100\,cm^{-1}$, which scales with the $H_2$ content, (iii) the isotopic data indicating a preferred partitioning of $^1H$ into the fluid during H loss. Still, further experimental work is needed to constrain the speciation and mobility of $H_2$ in mantle minerals and test the model presented here.

## Methods

### Water content and stable H isotopes determination

*"On-line" procedure.* Samples were analysed using a continuous flow elemental analyser (TC/EA) operating on-line with mass spectrometer[19,20,31]. The system used is a ThermoFisher HTFlash IRMS© working on-line with DeltaV+© mass spectrometer monitored by ConflowIV© diluter hosted at the Magmas and Volcanoes laboratory from Université Jean Monnet, Saint-Etienne. The DeltaV+© used an electrostatic filter to prevent isobaric interferences between the helium carrier gas and the generated mass 3 hydrogen. The elemental analyses used the pyrolysis line consisting of a glassy carbon tube filled with glassy carbon grains placed inside an alumina ceramic tube heated at 1450 °C and flushed by helium (100 ml min$^{-1}$). All hydrogenous gasses were reduced by glassy carbon, $H_2$ was separated from other gas species (CO) in a chromatographic column heated at 90 °C and transferred to mass spectrometer. ConflowIV© diluter monitored the flux and the two injections of $H_2$ reference gas manufactured by Air Liquid company. The duration of a complete analysis was 300 s. Aliquots of minerals weighting between 0.3 and 25 mg, depending on $H_2O$ content, were analysed. Samples were crushed to fine grains as suggested by[31,62] to prevent incomplete extraction and fractionation of H and D. All samples were preheated at 100 °C for 24 h to eliminate adsorption water on sample surfaces.

To estimate the impact of tiny fluid inclusions on water content and the role of molecular $H_2$ on ∂D, four samples experimented heating. 400 mg of pure handpicked omphacite with grain size ranging between 500 and 1000 μm were put in a 6 mm tube of fused quartz and connected to a vacuum preparation line ($10^{-3}$ mbar). Each sample was held under vacuum and heated with a heat gun by steps of 20 min at 250, 400, 500, and 600 °C. Between each step, an aliquot is taken for FTIR and TC/EA-IRMS analyses.

The relation between hydrogen contents and peak area detected by mass spectrometer was calibrated with benzoic acid (4.952 wt% H), and water concentrations were determined by mass $H_2$ peak area, the uncertainty is estimated to be ±0.05 wt%.

We have also investigated/addressed linearity issues by loading 36 aliquots of Biotite NBS30 of different weights (0.318–3.423 mg) and thus obtaining different peak sizes on mass 2 (amplitude) in the range of 741 mV to 9396 mV. At the beginning of each analytical session we applied the $H_3^+$ correction factor at different pressures of the reference gas to correct for different peak heights.

*D/H* measurements were calibrated against NBS30 biotite and IAEA CH7 polyethylene and previously measured amphibole and mica on VSMOW-GISP isotope scale[63] by modified off-line method of[64] (see below). These have been chosen based on their extremely different ∂D values (Amphibole AJE 282: ∂D = −130 ± 0.5‰ and Mica AJE361 ∂D = −40 ± 3‰ vs VSMOW) spanning over a range of 90‰ and overlapping with the range of unknown samples. A mean ∂D value for the NBS30 biotite standard of −65.4 ± 1‰ and water content of 3.67 ± 0.12 wt% (n = 36) were obtained during the course of this study.

*"Off-line" procedure.* A suite of amphibole (richterite) and mica (phlogopite) from the South African MARID suite was first analysed using an "off-line" vacuum extraction line. 30–80 mg of pure hand-picked mica and amphibole with grain size ranging between 100 and 200 μm were put in a 6 mm tube of fused quartz and connected to a vacuum preparation line ($10^{-9}$ mbar). Each sample was held at 150–200 °C for 1 h under vacuum to desorb atmospheric water, and heated gradually with a butane-oxygen torch to release all hydrogenous gas to reach the melting point of quartz tube (1700–1800 °C). On the line, a CuO grain furnace constantly held at 575 °C allowed to transform all hydrogenous gas to $H_2O$ that was subsequently collected at liquid nitrogen temperature in a 10 mm pyrex cold finger. The trap temperature was increased to −90 °C with a mixture of ethanol-liquid nitrogen to allow the non-condensable gases to be pumped away. Then the collected $H_2O$ was reduced to $H_2$ with U metal at 800 °C[64]. $H_2$ was trapped into a coconut charcoal cold finger at liquid nitrogen temperature and expanded at room temperature in a calibrated volume connected to a capacitance gauge allowing to measure the "total water" content of minerals. On the same line, water standards (IAEA VSMOW, GISP and lab standards) were converted to $H_2$ with the same procedure. The D/H ratios were determined using an Elementar Isoprime dual-inlet mass spectrometer at the Magmas and Volcanoes laboratory, Jean Monnet University, Saint-Etienne. The results are expressed in the ∂-notation as permil relative to VSMOW. A mean ∂D value for the IAEA NBS30 biotite standard of −65.7 ± 0.3‰ and water content of 3.68 ± 0.1 wt% (n = 4) were obtained during the course of this study.

**FTIR.** Ten grains of each sample were doubly polished with final thicknesses of 150–450 μm depending on the grains. The OH content has been determined on a Bruker Vertex 70 FTIR (Fourier transform infra-red spectroscope) coupled with a Hyperion microscope equipped with ×15 objective and condenser at LMV. Beam size in the analyses varied from 30 to 50 μm. The spectra were collected through a $CaF_2$ plate with a resolution of 2 cm$^{-1}$ and with up to 300 scans. After the application of a linear baseline with anchor points outside the OH stretching region, the absorbance was integrated from 3000 to 3800 cm$^{-1}$ and the absorbance coefficient for omphacite was applied[34]. The calculation of the water concentration was performed using the Beer-Lambert law: $A = \varepsilon \cdot C \cdot t$, where $A$ is the absorbance, $\varepsilon$ is the absorptivity, $C$ the concentration and $t$ the thickness (in cm). Quantification was based on the average of ~10 unpolarised measurements performed on randomly oriented grains within the doubly polished thin sections. The absolute absorbance of the crystal is then equal to three times the unpolarised value as demonstrated by[65]. Absorbances of molecular $H_2$ and $H_2O$ followed the same procedure and were integrated respectively from 4000 to 4300 cm$^{-1}$ and around 5200 cm$^{-1}$. The concentration of molecular water was calculated using previously published absorptivity[28]. The absorptivity of $H_2$ was calculated using the Beer-Lambert law and the concentrations calculated in Table 1 from the difference

between the total water content measured by TC-EA-IRMS and the water content measured by FTIR in the OH+H$_2$O frequency region (see Supplementary Fig. 1).

**SIMS**. In situ water contents were measured on polished sections, gold coated, with the Cameca IMS1280 HR ion microprobe at CRPG-CNRS, Nancy. A 13 kV, 5 nA O- primary beam was focused onto the sample to a diameter of 20 μm. The secondary beam mass resolution was set at 1600, with an energy window of 35 eV and no energy filtering. Secondary ions of H$^+$ and D$^+$ were measured by peak switching for 10 min by ion counting. Under these analytical conditions, counting rates on H+ varied between $1 \times 10^5$ and $5 \times 10^5$ counts per second and statistical precision ranged from 0.5 to 3%. Samples were carefully degassed before introduction in the analytical chamber. The samples were doubly polished thin sections that were glued on a glass plate and gold coated. They were introduced in the vacuum chamber of the SIMS the night prior to analysis at 2 μPa ($2 \times 10^{-9}$ atm) associated with a liquid N$_2$ cold trap. A presputering of 3 min with a 20 μm raster was used to clean the sample surface before measurement, and a raster of 5 μm and an electronic gate of 90% was used for the analysis. The background level was lower than 10 ppm of water. The water content of samples was calculated by comparing the measured hydrogen secondary ion intensity relative to the primary ion beam intensity of samples with that of pyroxene of known composition[66] measured during the same session as reference material. The estimated precision on the calculated water content was about 15% (1 sigma). All hydrogen signal is converted into water content, without considering its initial form.

**2-Pressure–temperature estimates**. Temperatures were calculated with a pressure-dependent garnet-clinopyroxene Fe–Mg geothermometer[67] for 1, 3, and 7 GPa. Pressures were calculated by projecting the temperature estimates to local conductive model geotherms[68] corresponding to a surface heat flow of 39 mW m$^{-2}$ for Roberts Victor[69] and of 45 mW m$^{-2}$ for Obnazhennaya (estimated at 40 to >50 mW m$^{-2}$)[70].

## Data availability
All data in this study are presented in Table 1 and available in Supplementary Tables 1 and 2.

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

## Acknowledgements

This research was supported by the Laboratory of Excellence ClerVolc, Jean Monnet University-Saint-Etienne and CPER AURA. AK, AG, and OO were supported by Russian Federation state assignment projects of IGM SB RAS and of DPMGI SB RAS. This research was financed by the French Government Laboratory of Excellence Initiative n° ANR-10-LABX-0006 and by the TelluS programme of CNRS/INSU. This is Laboratory of Excellence ClerVolc contribution n°415. The authors thank Prof. Chris Harris for discussions and his improvements on the English writing.

## Author contributions

B.N.M. performed the H$_2$O$_{tot}$ and ∂D measurements (TC/EA-IRMS and off-line/Dual-inlet method). I.B.R., B.N.M., and N.B-C. performed FTIR measurements, and E.D. performed SIMS measurements. I.B.R. calculated pressure–temperature equilibration conditions. G.C., D.I., A.G., A.K., and O.O. provided the samples. B.N.M., N.B-C. I.B.R., D.I., J-Y.C., A.K., A.G., and E.D contributed ideas, models, plots, and participated in writing of the manuscript. B.N.M. took the lead in preparing the manuscript with input from the other authors.

## Competing interests

The authors declare that they have no known competing financial interests or personal relationships that could have appeared to influence the work reported in this paper
