## [Peer Review File · Nature Communications]

Reviewers' comments:

Reviewer #1 (Remarks to the Author):

In this manuscript, the authors have conducted studies on the species, contents and isotopic compositions of H in omphacite of a suite of eclogites, by TCEA-IRMS, FTIR and SIMS. The authors reported the presence of molecular H₂ in natural minerals, confirming the experimental work of Yang et al. (2016) that molecular H₂ can be stored in appreciable amounts in nominally anhydrous minerals in the deep reduced mantle. This is important for insights into the deep H cycle.

I have some major concerns about the discussion and quantification of H₂.

1. Fractures are often well developed in mineral grains of eclogites, as shown in Figure 1 of this work and documented extensively elsewhere. It is very likely that tiny fluid inclusions (e.g., on nanoscales) are present along the fractures. Tiny fluid inclusions are also very common in normal eclogite minerals (e.g., regions free of fractures or fractures healed). These may have introduced in the peak metamorphism of eclogite and the subsequent processes. FTIR studies are in situ conducted on optically clean and fracture-free areas of well selected mineral grains — by this, it is possible that sometimes fluid inclusions in the sample is not probed. However, TCEA-IRMS and SIMS are very different, because what they measure is all H in a bulk sample (note that tiny fluid inclusions can be sealed in a sample and are not easy to be driven out of the sample by a routine heating). This may be the reason why you see very weak 5200 cm⁻¹ peak related to fluid water in your FTIR spectrum (Figure 2) but very high water contents by TCEA-IRMS and SIMS. Have you ever compared the FTIR spectra taken between optically clean and milky (dark and fractures developed) regions? Are they of similar 5200 cm⁻¹ peak?
2. The combined vibration of structural H in the lattice (e.g., 1600 + 3600 cm⁻¹) may cause a peak at ~5200 cm⁻¹. To observe this, sometimes you need thick samples. How thick are the samples (you should offer information here, which is also useful for a better knowledge of molecular H₂ in your samples). Another issue that should be clarified in the text: can the vibration of structural H in the lattice and non-structural fluid inclusions produce a combined peak at ~5200 cm⁻¹?
3. The IR bands at 3800-3000 cm⁻¹ (Figure 2) appear not typical of OH in omphacite. The broad band at ~3400 cm⁻¹ is probably due to tiny fluid inclusions. This peak contributes greatly to the amount of H at 3800-3000 cm⁻¹ (FTIR content). Did you include the inclusion water to your FTIR contents? The FTIR water contents are up to >1000 ppm H₂O, much higher than the OH solubility of omphacite in previous reports and also the OH contents of omphacite in eclogite xenoliths in available work. Could this be related to the tiny fluid inclusions in your samples?
4. Given the above, you cannot simply estimate the amount of H₂ in your samples by subtracting the FTIR contents from your TCEA-IRMS/SIMS values (what the latter has measured is not only OH plus H₂ but also huge amounts of inclusion water). Clarification of this issue (considering the complicated sources of inclusion water) may also help to discuss the D/H ratios.
5. The beginning paragraph of the text on page 2-3 may be restructured. Lines 44-46 should be rewritten, because Yang et al. (2016) have already demonstrated that H₂ could be stored in the deep reduced mantle and OH groups are not the only form of H in the mantle.
6. A minor issue is the terminology of the studied mineral (omphacite). You have used different names for it, e.g., pyroxenes of line 27 and omphacite and clinopyroxene in other places of text. A general name may be more suitable; moreover, pyroxenes usually denote two or more pyroxene-group minerals.

Reviewer #2 (Remarks to the Author):

This manuscript proposes the first evidence of molecular hydrogen in cpx recovered from eclogites. Evidence of H₂ in such minerals would be interesting as it could suggest H being present in the mantle as H₂ and not OH in minerals.

Despite this interesting topic, this manuscript suffers of one major flaw that prevents its publication in this journal: the infrared assignment is wrong, and, as everything is based on that, all the subsequent reasoning should be revised.

Indeed, the author based their analysis on the claim that the IR peak at approximately 4070 cm⁻¹, which extends between 3850 and 4230 cm⁻¹ (see figure 2), can be assigned to H₂. I'm sorry but they are very wrong there. The H₂ peak can indeed be IR active, see the publication of Schmidt et al., *Journal of Non-Crystalline Solids* 240 (1998) 91-103 for an example of H₂ in silica. However, looking at the latter publication, their figure 2 indicates that H₂ in silica glass gives a peak centered at 4140 cm⁻¹, and extending between 4100 and 4200 cm⁻¹. It is known that molecules in glasses yield broader spectra, like the broad signals of O-H stretching in glasses compared to minerals. Accordingly, the H₂ signals found by Yang et al. in NAMs are located at slightly lower frequency, around 4060 cm⁻¹, but they are narrower as expected, extending between 4030 and 4100 cm⁻¹ MAXIMUM.

This is why the peak found by the authors in this study, which spans nearly 380 cm⁻¹, cannot be assigned to H₂ without further rock-solid evidence (like a molecular dynamic study for instance). H₂ is a small molecule, and, even strongly perturbed by surrounding interactions as the authors infer (how do you do that by the way for a very small neutral molecule?), it is not reasonable to think that H₂ can give such a broad signal as such span in frequency will imply a non-realistic extent of the H-H bond distance. Now, turning to the real origin of this signal, I suggest the authors to look at the publication of Stolper 1982, *CMP*81:1 for water in glasses. In glasses, a broad signal around 4100 cm⁻¹ is always observed, and is assigned to some X-O-H combination mode with X some metal cations; this mode, not fully resolved at the moment in glasses, is also observed in argiles and other minerals. I think the authors are facing a similar vibrational mode there, and it surely cannot be assigned to H₂ in the case of their spectra.

Given such considerations, this manuscript should not be published at all and the authors should strongly revise their data and arguments.

Reviewer #3 (Remarks to the Author):

This study reports the first direct observation of molecular H₂ in natural mantle minerals. Moreover, the observed concentrations are amazingly high and appear to correlate well with hydrogen isotopic composition. As such, this is a very important study, which should be published after some minor revisions. A few style corrections by a native English speaker before resubmission would be desirable.

An absolutely critical point is the identification of the IR band at 4070 cm⁻¹ as being due to H₂. Here, I would recommend showing a plot (maybe in the supplementary material) of the absorbance (normalized to 1 cm thickness) of this peak versus the inferred H₂ content. This should give a straight line passing through the origin of the diagram. Another thing that could easily be done – to convince people that this is not an overtone of the omphacite lattice vibrations – would be some simple stepwise heating experiments with one omphacite sample and taking FTIR spectra after each step. If the peak is due to H₂, it should disappear already after minor heating.

Minor comments:

Line 27 ff: "pyroxenescontain molecular H₂ rather than water (dissolved as OH groups) as previously thought" – misleading statement, as these samples contain H₂ in addition to OH groups

Line 44: "estimated to amount, as water, up to ~7.4 times the mass of the oceans" – this is an extreme estimate; most studies are more consistent with 0.5 – 1 ocean mass.

Line 47: "high-pressure olivine variety found in a deep diamond" should be "ringwoodite inclusion in an ultradeep diamond from the transition zone"

Line 52: "Low optical activity" should be "low infrared extinction coefficient"; "optical activity" has a very specific meaning; it describes the ability of a substance to rotate the plane of linear polarized light – something completely different.

Line 102: "very low reactivity of H₂ to infrared radiation" – there is no "reactivity" – "very low infrared absorbance of H₂"

Line 115: The diffusivity of H⁺ in NAM's cannot be used as argument for the preservation of H₂, since H₂ is a different diffusing species, which will have completely different diffusion coefficients.

Line 187: "This must be taken into account when interpreting the H isotopic distribution in mantle and models involving deep mantle degassing" – one would think that degassing never happens in the deep mantle, but only rather close to the surface, as both mantle minerals and mantle melts are typically undersaturated with volatiles, except at very low pressures (< 1 GPa).

Figure 2: Presumably, this is an unpolarized infrared spectrum – this should be stated in the caption.

In Supplementary Table 1, subscripts, e.g. in SiO₂, Al₂O₃ are missing

Bayreuth, October 8, 2019

Hans Keppler

Response to Reviewers' comments:

-In red our answers and in yellow the text that was added to the manuscript-

Reviewer #1

In this manuscript, the authors have conducted studies on the species, contents and isotopic compositions of H in omphacite of a suite of eclogites, by TCEA-IRMS, FTIR and SIMS. The authors reported the presence of molecular H₂ in natural minerals, confirming the experimental work of Yang et al. (2016) that molecular H₂ can be stored in appreciable amounts in nominally anhydrous minerals in the deep reduced mantle. **This is important for insights into the deep H cycle.**

I have some major concerns about the discussion and quantification of H₂.

1. Fractures are often well developed in mineral grains of eclogites, as shown in Figure 1 of this work and documented extensively elsewhere. It is very likely that tiny fluid inclusions (e.g., on nanoscales) are present along the fractures. Tiny fluid inclusions are also very common in normal eclogite minerals (e.g., regions free of fractures or fractures healed). These may have introduced in the peak metamorphism of eclogite and the subsequent processes. FTIR studies are in situ conducted on optically clean and fracture-free areas of well selected mineral grains -- by this, it is possible that sometimes fluid inclusions in the sample is not probed. However, TCEA-IRMS and SIMS are very different, because what they measure is all H in a bulk sample (note that tiny fluid inclusions can be sealed in a sample and are not easy to be driven out of the sample by a routine heating). This may be the reason why you see very weak 5200 cm⁻¹ peak related to fluid water in your FTIR spectrum (Figure 2) but very high water contents by TCEA-IRMS and SIMS. Have you ever compared the FTIR spectra taken between optically clean and milky (dark and fractures developed) regions? Are they of similar 5200 cm⁻¹ peak?

⇒ The reviewer is right. Even if we are very careful in selecting very clear grains, we cannot guarantee avoiding nanoscale fluid inclusions or cracks. However, Figure 1 shows that the grains are fractured, but not whitish or translucent. Also in Figure 2 we show the presence of molecular water, associated to a very noisy band at 5200 cm⁻¹ band in the FTIR spectra and integration of this band indicates the presence of less than 200-300 ppm wt H₂O (see **Table 1 and Table 2 supp**) that could be associated to tiny fluid inclusions within the grains analyzed by FTIR.

⇒ The reviewer is suggesting that the bulk method sees more fluid-inclusion water than FTIR because FTIR cannot sample a 2nd or 3rd dimension defect that is heterogeneously distributed. This is a very legitimate comment, but as we added to the text:

“Since SIMS is a microbeam technique (that measures all H atoms independently of their speciation), the fact that the water contents it provides agrees with the bulk water

measurements by TC-EA/IRMS indicates that the amount of H in occasional micro-inclusions and/or fractures is low or negligible”

⇒ We also added to the text: “To confirm the speciation of H, the samples were heated under vacuum (10^{-3} mbar) at 400°C, with sample Obn112 incrementally heated at 250, 400, 500 and 600°C. After heating, the samples were re-analysed with FTIR and TC-EA/IRMS. These stepwise experiments show that (1) the bulk water content and the isotopic composition of the samples do not change after heating at 400°C (see **Supp. Tab. 2**), which must have expelled any H associated to hypothetical nano-bubbles as observed in fluid inclusion studies³²; (2) incrementally heating sample Obn112 up to 600°C increases the δD value of extracted H₂O (this can be only explained by loss of a more negative component such as H₂) (see **Supp. Fig. 2**), along with a decrease in OH and H₂ concentrations in the omphacite (confirmed spectroscopically by the decrease in the bands at 2800-3600 cm⁻¹ and at 4100 cm⁻¹, see Tab. 1 Supp. Tab. 2 and **Figure 2**).”

2. The combined vibration of structural H in the lattice (e.g., 1600 + 3600 cm⁻¹) may cause a peak at ~5200 cm⁻¹. To observe this, sometimes you need thick samples. How thick are the samples (you should offer information here, which is also useful for a better knowledge of molecular H₂ in your samples).

⇒ We added the thickness of the samples in the Analytical technics section, sorry about that.

Another issue that should be clarified in the text: can the vibration of structural H in the lattice and non-structural fluid inclusions produce a combined peak at ~5200 cm⁻¹?

⇒ Usually this band is interpreted as molecular water, and anyways it is not critical since it is very weak. Considering the intensity of the 3600 and 1600 cm⁻¹ bands we expect that a combination of these two would be much more intense.

3. The IR bands at 3800-3000 cm⁻¹ (Figure 2) appear not typical of OH in omphacite.

⇒ This comment is confusing, as all previous IR studied on omphacites e.g. Huang et al. (2014) or also Koch-Mueller et al. (2004, 2008) show that bands between 3400 and 3600 cm⁻¹ are typical of omphacite, see Figures below.

The broad band at $\sim 3400\text{ cm}^{-1}$ is probably due to tiny fluid inclusions. This peak contributes greatly to the amount of H at $3800\text{--}3000\text{ cm}^{-1}$ (FTIR content). Did you include the inclusion water to your FTIR contents? The FTIR water contents are up to $>1000\text{ ppm H}_2\text{O}$, much higher than the OH solubility of omphacite in previous reports and also the OH contents of omphacite in eclogite xenoliths in available work. Could this be related to the tiny fluid inclusions in your samples?

⇒ Concerning the high water contents observed here, they are not unusual compared to previous reports: for example Huang et al (2014) find $1400\text{ ppm wt H}_2\text{O}$ in Robert Victor eclogites; Katayama and Nakashima (2003) recalculate up to $3000\text{ ppm wt H}_2\text{O}$ in eclogitic clinopyroxenes from Kokchetav, Kazakhstan. Also, by calculating the original composition of omphacite at high pressure, Smyth et al. (1991) derived a water content of $4000\text{ ppm wt. H}_2\text{O}$ in Robert Victor.

⇒ If molecular water is present as tiny fluid inclusions as suggested by the reviewer then it should also vibrate at 5200 cm^{-1} . However, this band is very noisy and does not correspond to more than 300 ppm by weight of water (see **Table 1 and Table 2 supp**). Moreover, the 3400 cm^{-1} band is not really characteristic of our infrared omphacite spectra. If the spectrum is deconvoluted, the band at 3400 cm^{-1} is present but absolutely not dominant.

[REDACTED]

Figure Typical spectrum for our omphacites (top) and Huang et al (2014) (bottom)

4. Given the above, you cannot simply estimate the amount of H₂ in your samples by subtracting the FTIR contents from your TCEA-IRMS/SIMS values (what the latter has measured is not only OH plus H₂ but also huge amounts of inclusion water). Clarification of this issue (considering the complicated sources of inclusion water) may also help to discuss the D/H ratios.

⇒ The area of the band at 5200 cm⁻¹ that reveals molecular water hence fluid inclusions corresponds to less than 300 ppm wt H₂O (see **Table 1 and Table 2 supp**). Also the bulk water content does not vary by heating at 400°C for 20 min. This proves that the amount of molecular water related to hypothetical nano-bubbles is negligible.

5. The beginning paragraph of the text on page 2-3 may be restructured. Lines 44-46 should be rewritten, because Yang et al. (2016) have already demonstrated that H₂ could be stored in the deep reduced mantle and OH groups are not the only form of H in the mantle.

⇒ We agree, and this is said so in our introduction “So far, hydrogen has been thought to exist in the mantle in the form of hydroxyl (OH), with storage capacity depending on depth. A few hundred ppm by weight were measured in peridotite xenoliths (mantle fragments brought up by volcanic eruptions) and up to 1.2 wt.% in a high-pressure olivine variety found in a deep diamond¹⁰. On the other hand, xenolith studies indicate that oxygen fugacity, f_{O_2} , decreases with increasing depth¹¹ such that at ≥ 250 km a substantial amount of H should be present in a reduced form¹². Reducing conditions were thought to greatly reduce the solubility

of OH in olivine¹³, for instance, however, it was recently discovered that H₂ could also be dissolved in NAMs, its low optical activity having hindered its detection for many years. »

6. A minor issue is the terminology of the studied mineral (omphacite). You have used different names for it, e.g., pyroxenes of line 27 and omphacite and clinopyroxene in other places of text. A general name may be more suitable; moreover, pyroxenes usually denote two or more pyroxene-group minerals.

=>we corrected that.

Reviewer #2

This manuscript proposes the first evidence of molecular hydrogen in cpx recovered from eclogites. Evidence of H₂ in such minerals would be interesting as it could suggest H being present in the mantle as H₂ and not OH in minerals.

Despite this interesting topic, this manuscript suffers of one major flaw that prevents its publication in this journal: the infrared assignment is wrong, and, as everything is based on that, all the subsequent reasoning should be revised.

Indeed, the author based their analysis on the claim that the IR peak at approximately 4070 cm⁻¹, which extends between 3850 and 4230 cm⁻¹ (see figure 2), can be assigned to H₂. I'm sorry but they are very wrong there. The H₂ peak can indeed be IR active, see the publication of Schmidt et al., Journal of Non-Crystalline Solids 240 (1998) 91-103 for an example of H₂ in silica. However, looking at the latter publication, their figure 2 indicates that H₂ in silica glass gives a peak centered at 4140 cm⁻¹, and extending between 4100 and 4200 cm⁻¹. It is known that molecules in glasses yield broader spectra, like the broad signals of O-H stretching in glasses compared to minerals. Accordingly, the H₂ signals found by Yang et al. in NAMs are located at slightly lower frequency, around 4060 cm⁻¹, but they are narrower as expected, extending between 4030 and 4100 cm⁻¹ MAXIMUM.

This is why the peak found by the authors in this study, which spans nearly 380 cm⁻¹, cannot be assigned to H₂ without further rock-solid evidence (like a molecular dynamic study for instance). H₂ is a small molecule, and, even strongly perturbed by surrounding interactions as the authors infer (how do you do that by the way for a very small neutral molecule?), it is not reasonable to think that H₂ can give such a broad signal as such span in frequency will imply a non-realistic extent of the H-H bond distance.

⇒ We believe there are more different ionic environments in omphacite than in silica glass, which essentially contains only silicon tetrahedra. We do not see a broad band but many overlapping contributions. The heated samples show very clearly a narrow peak at 4100cm⁻¹ corresponding to molecular H₂. We do see a large peak on unheated samples, but it is the result of (1) very high concentrations of H₂ (100 times higher than that detected by Yang et al 2016: 400-460ppm versus 3-5ppm, and we know that peaks widen with the increasing concentration), (2) pyroxenes may host H₂ in several ionic environments resulting in several possible peaks (4100 and 4200 cm⁻¹) which can overlap. To take this criticism into

account, however, we recalculated absorbance only between 4000 and 4300 cm^{-1} , thus eliminating the 3950 cm^{-1} peak described by Stolper 1982 and probably attributable to molecular water. (see extract from the Fig. 2 below)

Extract from the Figure 2: Unpolarized FTIR spectra of omphacite (Obn112).

Secondary peaks in the 3800 – 6000 cm^{-1} interval correspond to molecular H_2 (4100 cm^{-1}), Si-OH bending (4470 cm^{-1}) and molecular water (5200 cm^{-1}). Comparison between unheated and heated samples (500 and 600°C) evidences the decrease of intensity of the peak at 4100 cm^{-1} with increasing temperature. The widening of the peak with increasing H_2 content is due to overlapping with two smaller peaks at 4200 and 3950 cm^{-1} .

Now, turning to the real origin of this signal, I suggest the authors to look at the publication of Stolper 1982, CMP81:1 for water in glasses. In glasses, a broad signal around 4100 cm^{-1} is always observed, and is

assigned to some X-O-H combination mode with X some metal cations; this mode, not fully resolved at the moment in glasses, is also observed in argiles and other minerals. I think the authors are facing a similar vibrational mode there, and it surely cannot be assigned to H₂ in the case of their spectra.

⇒ If we understand correctly the reviewer suggests that the bands observed around 4100 cm⁻¹ are not due to H₂? If the band at 4100 cm⁻¹ in glasses is due to X-OH combination mode as the reviewer claims (X being a cation other than H) why is it also observed in water and ice (where there is no X) ? By the way Stolper himself did not attribute the 4100 cm⁻¹ band to some X-O-H combination mode. So if this assignment has not been published/resolved for glasses why should we trust it to be present in our samples and with such a low intensity? Besides we do observe a band at 4470 cm⁻¹ that we assign to Si-OH bending.

Reviewer #3

This study reports the first direct observation of molecular H₂ in natural mantle minerals. Moreover, the observed concentrations are amazingly high and appear to correlate well with hydrogen isotopic composition. As such, this is a very important study, which should be published after some minor revisions. A few style corrections by a native English speaker before resubmission would be desirable.

An absolutely critical point is the identification of the IR band at 4070 cm⁻¹ as being due to H₂. Here, I would recommend showing a plot (maybe in the supplementary material) of the absorbance (normalized to 1 cm thickness) of this peak versus the inferred H₂ content. This should give a straight line passing through the origin of the diagram.

⇒ With the exception of one sample (Obn110) all samples align very well on a straight line passing through zero in a diagram of the H₂ concentration versus the absorbance (see **Supp. Fig.1**). This demonstrates that the 4100 cm⁻¹ band is indeed due to H₂ and not an overtone the silicate lattice vibration.

Supp.Figure.1: Calculated molecular H₂ versus Integrated Absorbance normalized to 1cm corresponding to 4100 cm⁻¹ peak. With the exception of one sample (Obn110) all samples align very well on a straight line passing through zero in a diagram of the H₂ concentration versus the absorbance. This demonstrates that the 4100 cm⁻¹ band is indeed due to H₂ and not an overtone of the silicate lattice vibration.

Another thing that could easily be done - to convince people that this is not an overtone of the omphacite lattice vibrations - would be some simple stepwise heating experiments with one omphacite sample and taking FTIR spectra after each step. If the peak is due to H₂, it should disappear already after minor heating.

=> We do observe that after heating the intensity of the H₂ peak decreases, which means that it is not due to an overtone of the silicate vibration. We added to the text: **“To confirm the speciation of H, the samples were heated under vacuum (10⁻³ mbar) at 400°C, with sample Obn112 incrementally heated at 250, 400, 500 and 600°C. After heating, the samples were re-analysed with FTIR and TC-EA/IRMS. These stepwise experiments show that (1) the bulk water content and the isotopic composition of the samples do not change after heating at 400°C (see **Supp. Tab. 2**), which must have expelled any H associated to hypothetical nano-bubbles as observed in fluid inclusion studies³²; (2) incrementally heating sample Obn112 up to 600°C increases the δD value of extracted H₂O (this can be only explained by loss of a more negative component such as H₂)(see **Supp. Fig. 2**), along with a decrease in OH and H₂ concentrations in the omphacite**

(confirmed spectroscopically by the decrease in the bands at 2800-3600 cm^{-1} and at 4100 cm^{-1} , see Tab. 1 Supp. Tab. 2 and Figure 2).”

Supp.Figure2: δD versus H_2O (TCEA-IRMS) of the heated and unheated omphacite Obn112. Total water content and δD are well correlated. Here, the loss of H implies heavier isotopic compositions. This is due to a loss of much more negative component (H_2 rather than OH or H_2O), which is also observed in infrared spectra where the integrated absorbances decrease significantly with the heating temperature.

Supp.Fig.2: The increase in δD with bulk water loss upon heating is concomitant with a loss of OH and H_2 from the sample. Such behavior can be well explained by a positive fractionation factors between the mineral and the fluid/vapor phase ($\text{H}_2\text{O}-\text{H}_2$)

Minor comments:

Line 27 ff: “pyroxenes ...contain molecular H_2 rather than water (dissolved as OH groups) as previously thought” – misleading statement, as these samples contain H_2 in addition to OH groups.

⇒ Indeed, we rephrased to: “ Here we show for the first time the presence of H_2 in natural minerals, found in eclogites (basaltic rocks recycled back into the mantle) from the Kaapvaal and Siberian cratons. In these omphacites high levels of H_2 (from 70 to 460 ppm wt H_2) are found to coexist with OH.”

Line 44: "estimated to amount, as water, up to ~7.4 times the mass of the oceans" - this is an extreme estimate; most studies are more consistent with 0.5 - 1 ocean mass.

⇒ We agree of course. We modified: "...estimated to amount, as water, to 0.5-1 times the mass of the oceans^{4,5}, with cosmochemical arguments leading to an original estimate in the Bulk Silicate Earth of ~7 oceanic masses⁶ ».

Line 47: "high-pressure olivine variety found in a deep diamond" should be "ringwoodite inclusion in an ultradeep diamond from the transition zone"

⇒ Ok with that. We changed the text accordingly.

Line 52: "Low optical activity" should be "low infrared extinction coefficient"; "optical activity" has a very specific meaning; it describes the ability of a substance to rotate the plane of linear polarized light - something completely different.

⇒ Ok with that. We changed the text accordingly.

Line 102: "very low reactivity of H₂ to infrared radiation" - there is no "reactivity" - "very low infrared absorbance of H₂"

⇒ Ok with that. We changed the text accordingly.

Line 115: The diffusivity of H⁺ in NAM's cannot be used as argument for the preservation of H₂, since H₂ is a different diffusing species, which will have completely different diffusion coefficients.

That is correct. We clarified our thoughts in the text: "However, the preservation within the mantle of high amounts of H (> 2000 ppm), considered highly mobile, at high temperatures and for a long time, needs to be explained. Diffusivity of H₂ is currently unknown, but a lower limit could be that of H⁺, which is a much smaller speciation/configuration. The diffusion rate of H⁺ in NAMS under mantle conditions is high, between 10⁻⁸ and 10⁻¹⁴ m²s⁻¹ equivalent to 120 m/Ga⁽³⁴⁾ at 1100°C. Molecular hydrogen is larger than ionic H⁺, with an ionic size of 75 pm, which is as large as Mg²⁺ (72 pm). Accordingly, H₂ may diffuse much slower than H⁺. Such diffusivity implies that equilibrium is geologically fast at the mineral grain scale, yet high H concentrations could be maintained on the scale of oceanic crust fragments hundreds of metres or kilometres in size. If such eclogitic blocks are preserved for several Ga in the mantle, they will develop zoning of H in terms of abundance and isotope ratio, reflecting their progressive re-equilibration with the ambient mantle. "

Line 187: "This must be taken into account when interpreting the H isotopic distribution in mantle and models involving deep mantle degassing" - one

would think that degassing never happens in the deep mantle, but only rather close to the surface, as both mantle minerals and mantle melts are typically undersaturated with volatiles, except at very low pressures (< 1 GPa).

⇒ Instead of degassing we can say "loss of volatiles".

Figure 2: Presumably, this is an unpolarized infrared spectrum - this should be stated in the caption.

⇒ Ok, we specified in the Figure caption.

In Supplementary Table 1, subscripts, e.g. in SiO₂, Al₂O₃ are missing

⇒ Ok, done.

Bayreuth, October 8, 2019

Hans Keppler

Reviewers' comments:

Reviewer #1 (Remarks to the Author):

In the revised manuscript, the authors have provided new results of heating experiments, as well as re-writings of the text. As said, the results are important for insights into the deep H cycle -- but this is true only when (1) what you have seen in the spectra is indeed due to molecular hydrogen and (2) the quantification (the different H contents between SIMS/TCEA accounted for by molecular hydrogen) is correct. The current form of the manuscript is, however, not convincing. In addition to issues already mentioned, there are new problems arising. If these issues, not limited to the following ones, are addressed convincingly and carefully, the work may make a nice contribution.

1. The heating experiments. The mobility of molecular hydrogen is extremely fast and its loss from a host mineral or glass is very easy upon very minor heating and even at room temperature. You show the reduced intensity of the 4070 cm⁻¹ band at 500 and 600 oC: did you see any apparent intensity loss at lower temperature, and how long have you heated the sample at 500/600 oC? In your Fig. 2, the 4470 cm⁻¹ band, as assigned to the bending vibration of OH, demonstrates similar loss of band intensity upon heating. This means that your sample lost OH at the same time, which makes things complicated. If your 4070 cm⁻¹ band was indeed caused by molecular hydrogen, it would be better if you heat your sample "very slightly" at very low temperatures and with very short durations, so that you see only the loss of "molecular hydrogen" (if it were indeed molecular hydrogen) while the intensity of OH is maintained.

2. Given that molecular hydrogen is lost easily even at room temperature (as indicated by many available reports), it is surprising to see that your samples have preserved so much "molecular hydrogen". What is the formation age of your samples and how long have they been exhumed and exposed at the surface? For a 1 mm sized glass or mineral, the complete loss of molecular hydrogen at room temperature may be only a few to tens of Ma... For SIMS work, samples are usually baked under vacuum and at >100 oC for >20 hours prior to analyses (similar procedure for TCEA work?). Why was "molecular hydrogen" not lost from your sample during that pre-heating? If the loss was unavoidable, can you use the SIMS/TCEA analyses to quantify your "molecular hydrogen" contents?

3. Omphacites in natural eclogites usually show OH bands at ~3630-3610, 3530-3510 and 3460-3440 cm⁻¹, and the band at ~3450 cm⁻¹ is the strongest (as documented extensively elsewhere). The spectra you have shown are not typical of OH in natural omphacites (even for your spectra you noted in your rebuttal). I am still not convinced that fluid inclusions contribute very little to your bulk H contents measured by SIMS/TCEA (in particular if you consider the above mentioned behaviors/property of molecular hydrogen; also, it is obvious that, for in situ FTIR study, one easily chooses regions free of fractures and visible inclusions, which are hard for SIMS/TCEA work). Fluid inclusions are so common in natural omphacites (many reports are available on this issue, e.g., Vallis & Scambelluri, 1996, *Lithos*, 39, 81-92 and Koch-Muller et al., 2004, *American Mineralogist*, 89, 921-931), and their contribution may not be that small (not sure about your samples concerning the distribution of inclusions).

4. In your rebuttal, you argued that ionic environments of the host minerals affect the "molecular hydrogen" band (e.g. the band width). However, the results of Yang et al. (2016) have shown that the band of molecular hydrogen is broadly independent of the structure and composition of the host materials.

Xiaozhi Yang

Reviewer #2 (Remarks to the Author):

The authors have addressed well the points from the different reviewers, and the manuscript seems nearly ready for publication.

However, regarding the assignment of the H₂ peak, a final proof could be added.

Indeed, after reading a few references about H₂ in MOFs (see Nijem et al., [dx.doi.org/10.1021/ja2010863](https://doi.org/10.1021/ja2010863) and [dx.doi.org/10.1021/ja104923f](https://doi.org/10.1021/ja104923f)) it seems that H₂ signals in minerals may be broad in some cases, comforting the authors attribution, but this is not really the general case. The correlation with calculated H₂ contents is nice, but could always be discussed as the H₂ content is not measured by an independent method and only calculated.

The way to fully confirm that the authors are observing a H₂ band and not any OH-lattice combination mode is to perform Raman spectroscopy, because OH lattice and combination modes are Raman inactive, while the H₂ signal is Raman active.

While Raman usually lacks of sensitivity, the H₂ signal should be visible in the Raman spectrum of the mineral with 500 ppm H₂.

If the authors could bring this proof, this will fully clear any doubt regarding the assignment of the H₂ peak, making their manuscript rock-solid.

Reviewer #3 (Remarks to the Author):

This manuscript has been greatly improved upon revision and it could now be published with very minor changes. In particular, the authors have carried out additional experiments which support the assignment of the 4100 cm⁻¹ band to molecular hydrogen H₂: (1) Figure 2 shows that upon heating, the 4100 cm⁻¹ band assigned to H₂ decreases more strongly than the 4500 cm⁻¹ OH peak, as expected for the faster diffusivity of H₂; (2) Supplementary Figure 2 also shows that the 4100 cm⁻¹ band intensity is proportional to the difference between total water content and water dissolved as OH.

Taken together, these observations – together with the isotopic fractionation effects observed - strongly support the presence of molecular H₂ in the samples.

There are some minor aspects that could still be improved.

The results of the heating experiments are not well presented in lines 112 to 122. The most important point here is that the 4100 cm⁻¹ H₂ band behaves differently than the 4500 cm⁻¹ OH band, confirming that these bands are caused by two different species. I would therefore reword line 120 ff: "along with a strong decrease of H₂ concentrations and a minor decrease of OH in the omphacite as inferred spectroscopically from the intensity of the bands at 4100 cm⁻¹ and at 2800-3600 cm⁻¹ (Figure 2)."

The discussion of H₂ diffusion based on the diffusion of H⁺ in line 130 ff is unconvincing. Yes, H₂ is larger than H⁺, but H⁺ is a charged species, such that the diffusion of H⁺ requires a counter diffusion of some other charged species to maintain local charge balance. In contrast to this, H₂ can diffuse alone with little interaction with the matrix. The mobility of H₂ should be discussed on

the basis of H₂ diffusion data; while there are no such data for omphacite, there are data for silica glass, which may at least give some reasonable idea about the expected order of magnitude of the H₂ diffusion coefficient: Lee RW (1963): Diffusion of Hydrogen in Natural and Synthetic Fused Quartz. J. Chem. Phys. 38: 448.

In Figure 3 A, horizontal axis, "H₂O content" should be "total water content" for clarity.

In Table 1 and also in Supplementary Table 2, it is important to clarify that ppm is actually "ppm by weight" and ppm H₂ refers to "ppm by weight of H₂", not to ppm by weight of the equivalent amount of H₂O (as in many other studies).

At least in my pdf file, Supplementary Table S1 (Major element compositions) appears to be missing; there is only the caption, not the table. But Table 1 from the main article has been copied in again at the end of the supplementary material.

The English of the manuscript is generally understandable, but some minor corrections by a native English speaker would be desirable.

Bayreuth, January 22, 2020

Hans Keppler

Dr. Bertrand N. Moine
Lab. Magmas et Volcans
UMR6524 CNRS
Université J. Monnet
23 rue du Dr. P. Michelon
42023 Saint-Etienne Cedex 02
FRANCE

Response to Reviewers' comments:

Reviewers' comments:

Reviewer #1:

In the revised manuscript, the authors have provided new results of heating experiments, as well as re-writings of the text. As said, the results are important for insights into the deep H cycle -- but this is true only when (1) what you have seen in the spectra is indeed due to molecular hydrogen and (2) the quantification (the different H contents between SIMS/TCEA accounted for by molecular hydrogen) is correct. The current form of the manuscript is, however, not convincing. In addition to issues already mentioned, there are new problems arising. If these issues, not limited to the following ones, are addressed convincingly and carefully, the work may make a nice contribution.

1. The heating experiments. **The mobility of molecular hydrogen is extremely fast** and its loss from a host mineral or glass is very easy upon very minor heating and even at room temperature. You show the reduced intensity of the 4070 cm⁻¹ band at 500 and 600 oC: did you see any apparent intensity loss at lower temperature, and how long have you heated the sample at 500/600 oC?

⇒ *In fact we do not observe a loss of H₂ at low-T, we added L. 142: "As shown in Figure 2, the absorbance of the 4100 cm⁻¹ band decreases with increasing temperatures, and this decrease is decoupled from that of the band at ~4500 cm⁻¹, implying that these two bands cannot be attributed to the same species, i.e. to some combination of OH mode. Furthermore, the rate of the absorbance increase for the H₂ band is higher at low temperatures and becomes lower than for OH at temperatures ≥ 400°C (see Suppl. Figure 2 & 3). This indicates that the kinetics of H₂ and H⁺ diffusion are close in this temperature range and cross over at higher temperatures in agreement with previous measurements (Figure 5). Thus, we interpret the contrasting behaviour shown in Suppl. Figure 2 & 3 as a transition from OH to H₂ during the heating stage at low temperatures following the oxidation-dehydrogenation shown in reaction 1, similar to what is inferred for natural or experimental samples^{14,38,39}."*

We already explained in the Methods, Line 600: "To estimate the impact of tiny fluid inclusions on water content and the role of molecular H₂ on δD, 4 samples experimented heating. 200 mg of pure handpicked omphacite with grain size ranging between 500-1000µm were put in a 6mm tube of fused quartz and connected to a vacuum preparation line (10⁻⁶ mbar). Each sample was held under

Dr. Bertrand N. Moine
Lab. Magmas et Volcans
UMR6524 CNRS
Université J. Monnet
23 rue du Dr. P. Michelon
42023 Saint-Etienne Cedex 02
FRANCE

vacuum and heated with a heat gun by step for **20 min** at 250, 400, 500 and 600°C. Between each step, an aliquot is taken for FTIR and TC/EA-IRMS analyses.”

We corrected the grain size because 200-400 μm was the size after polishing for FTIR analyses; the grain size during heating experiment was 500-1000 μm .

In your Fig. 2, the 4470 cm^{-1} band, as assigned to the bending vibration of OH, demonstrates similar loss of band intensity upon heating. This means that your sample lost OH at the same time, which makes things complicated. If your 4070 cm^{-1} band was indeed caused by molecular hydrogen, it would be better if you heat your sample “very slightly” at very low temperatures and with very short durations, so that you see only the loss of “molecular hydrogen” (if it were indeed molecular hydrogen) while the intensity of OH is maintained.

⇒ *Indeed OH loss occurs at the same time as H_2 loss. This is normal given that they have similar diffusivities (see below). As a matter of fact, each sample was held at 100°C for 24h to desorb water before analyses (as already specified in the Methods section **L.598**) prior to the TC-EA-IRMS analysis. No difference in bulk water content was observed between the analysis performed with TC-EA-IRMS and that performed with SIMS, as shown for sample Obn110 thus it means that the 100°C heating stage did not affect the H bulk concentration. Indeed, according to the diffusivity of H_2 (Lee, 1963 & Shang et al., 2009) the loss of H_2 at 100°C should be negligible or within the error bar for grains of 500-1000 μm .*

2. Given that molecular hydrogen is lost easily even at room temperature (as indicated by many available reports), it is surprising to see that your samples have preserved so much “molecular hydrogen”. What is the formation age of your samples and how long have they been exhumed and exposed at the surface? For a 1 mm sized glass or mineral, the complete loss of molecular hydrogen at room temperature may be only a few to tens of Ma...

⇒ *If we look at the diffusivity of H_2 in silica glass as suggested by Reviewers 1 and 3, we don't find it so fast. We added this to the text **L.162**: “For example, the diffusivity of H_2 in silica glass^{38,39} is $9.3 \times 10^{-16} \text{ m}^2 \cdot \text{s}^{-1}$ at 23°C and $2.4 \times 10^{-12} \text{ m}^2 \cdot \text{s}^{-1}$ at 250°C. Using the activation energy of 44 $\text{kJ} \cdot \text{mol}^{-1}$ provided in these studies we calculate a diffusivity of H_2 of $1.5 \times 10^{-10} \text{ m}^2 \cdot \text{s}^{-1}$ at 600°C or $1.0 \times 10^{-9} \text{ m}^2 \cdot \text{s}^{-1}$ at 1000°C (**Figure 5**). Such values are very similar to those determined for the rate of reaction⁴³ as well as for OH diffusivity in olivine³⁶ or diopside³⁷. It follows that at 20°C, H_2 diffuses over 1 mm in 1.075 Ga, thus for a xenolith of 10 cm radius, a complete dehydrogenation would require 10^4 Ga. The age of the kimberlitic*

Dr. Bertrand N. Moine
Lab. Magmas et Volcans
UMR6524 CNRS
Université J. Monnet
23 rue du Dr. P. Michelon
42023 Saint-Etienne Cedex 02
FRANCE

eruption is 160 Ma, so we should not expect a significant H₂ degassing of the xenolith stored at near-surface conditions.”

⇒ *Schmidt et al. (1998) note that they have lost 35% of the initial H₂ in silica glass after 2 months, while Hirschmann et al. (2012) did not observe any difference in concentrations after 4 months but they stored their samples at -2°C. Thus, given the nature of our samples, we have little control on their storage over time in the field.*

For SIMS work, samples are usually baked under vacuum and at >100 °C for >20 hours prior to analyses (similar procedure for TCEA work?). Why was “molecular hydrogen” not lost from your sample during that pre-heating? If the loss was unavoidable, can you use the SIMS/TCEA analyses to quantify your “molecular hydrogen” contents?

⇒ *“The samples were doubly polished thin sections that were glued on a glass plate and gold coated. They were introduced in the vacuum chamber of the SIMS the night prior to analysis at 2 μPa (2.10⁻⁹ atm.) associated with a liquid N₂ cold trap.” We added this specification in the Methods section? L.656. The samples were not heated neither before nor in the SIMS. We remind here that “A presputtering of 3 mn with a 20 μm raster was used to clean the sample surface before measurement to eliminate adsorbed water”.*

⇒ *Anyways, given the published diffusion coefficient of Shang et al. (2009) et Lee et al. (1963) and given that we do not observe a discrepancy between SIMS and TC-EA-IRMS analysis of unheated samples, we should not expect any loss of H₂ in the SIMS.*

3. Omphacites in natural eclogites usually show OH bands at ~3630-3610, 3530-3510 and 3460-3440 cm⁻¹, and the band at ~3450 cm⁻¹ is the strongest (as documented extensively elsewhere). The spectra you have shown are not typical of OH in natural omphacites (even for your spectra you noted in your rebuttal). **I am still not convinced that fluid inclusions contribute very little to your bulk H contents measured by SIMS/TCEA** (in particular if you consider the above mentioned behaviors/property of molecular hydrogen; also, it is obvious that, for in situ FTIR study, one easily chooses regions free of fractures and visible inclusions, which are hard for SIMS/TCEA work). **Fluid inclusions are so common in natural omphacites** (many reports are available on this issue, e.g., Vallis & Scambelluri, 1996, Lithos, 39, 81-92 and Koch-Muller et al., 2004, American Mineralogist, 89, 921-931), and their contribution may not be that small (not sure about your samples concerning the distribution of inclusions).

⇒ *We are not saying that we do not have fluid inclusions because we do see some fluid water associated to the band at 5200 cm⁻¹, still the contribution is very small as evidenced by the low intensity of this band.*

Dr. Bertrand N. Moine
Lab. Magmas et Volcans
UMR6524 CNRS
Université J. Monnet
23 rue du Dr. P. Michelon
42023 Saint-Etienne Cedex 02
FRANCE

- ⇒ *Also, we do not see the band at 3400 cm⁻¹ that Gong et al. (2007) and Sheng & Gong (2017) ascribe to molecular water, and indeed it decreases from the water of garnet and omphacite from eclogite of Dabie Sulu.*
- ⇒ *Secondly the omphacites of Valli and Scambelluri are alpine eclogites, therefore they experienced slow exhumation in the course of which the sample has time to exsolve water due to exsolution (see Scambelluri and Philippot, 2001*
- ⇒ *We added **line 135**: “Structural H₂ is indeed observed in omphacite, and the linear relationship between the absorbance of the 4100 cm⁻¹ band and the calculated H₂ content indicates that its quantification is robust. Molecular water present in nano-inclusions seems to be negligible in these samples. However, if present in large quantities nano-inclusions could also be filled with H₂ given that the conditions of equilibration of the present eclogites are very close to the conditions where H₂ and H₂O are miscible within the mantle ³⁶”*

4. In your rebuttal, you argued that ionic environments of the host minerals affect the “molecular hydrogen” band (e.g. the band width). However, the results of Yang et al. (2016) have shown that the band of molecular hydrogen is broadly independent of the structure and composition of the host materials.

- ⇒ *In the paper of Yang et al. (2016) the concentrations of H₂ are very low, so it is not surprising that we cannot observe shoulders of the H₂ peak. The position of the peak (4062 cm⁻¹) is indeed similar for the different minerals in the Yang et al.(2016)s’ study, but the symmetry of the peak disappears for clinopyroxene equilibrated at the highest pressure with the highest H₂ content.*

Reviewer #2:

The authors have addressed well the points from the different reviewers, and the manuscript seems nearly ready for publication.

However, regarding the assignment of the H₂ peak, a final proof could be added.

Indeed, after reading a few references about H₂ in MOFs (see Nijem et al., [dx.doi.org/10.1021/ja2010863](https://doi.org/10.1021/ja2010863) and [dx.doi.org/10.1021/ja104923f](https://doi.org/10.1021/ja104923f)) it seems that H₂ signals in minerals may be broad in some cases, comforting the authors attribution, but this is not really the general case. **The correlation with calculated H₂ contents is nice, but could always be discussed as the H₂ content is not measured by an independent method and only calculated.**

- ⇒ *“In minerals, H₂ has been identified before at a frequency of 4062 cm⁻¹ in the FTIR spectra (Yang et al., 2016). In glasses, the frequency of vibration of H₂ is located at higher wavenumbers of 4130 cm⁻¹ to 4140 cm⁻¹ and actually increases with the pressure of synthesis (Hirschmann et al., 2012).” So the identification of H₂ based on spectroscopy is robust.*

Dr. Bertrand N. Moine
Lab. Magmas et Volcans
UMR6524 CNRS
Université J. Monnet
23 rue du Dr. P. Michelon
42023 Saint-Etienne Cedex 02
FRANCE

⇒ *H₂ is evidenced by three observations:*

- 1- The measurement of H₂ species by FTIR spectroscopy, which has always been considered the most sensitive method for quantifying H in minerals.*
- 2- The discrepancy between TC-EA-IRMS and OH absorbance, associated to the low quantity of molecular water in tiny inclusions (as evidenced by a low intensity at 5200 cm⁻¹, lack of band at 3400 cm⁻¹), pointing to a missing species.*
- 3- The variation of the isotopic composition of H in these samples as a function of water content as well as the behavior of δD during heating, point out to the involvement of H₂.*

The way to fully confirm that the authors are observing a H₂ band and not any OH-lattice combination mode is to perform Raman spectroscopy, because OH lattice and combination modes are Raman inactive, while the H₂ signal is Raman active.

While Raman usually lacks of sensitivity, the H₂ signal should be visible in the Raman spectrum of the mineral with 500 ppm H₂. If the authors could bring this proof, this will fully clear any doubt regarding the assignment of the H₂ peak, making their manuscript rock-solid.

- ⇒ *We do not agree here. OH lattice modes can be Raman active as we have already quantified OH in olivine using Raman (Bolfan-Casanova et al., 2014).*
- ⇒ *As said by the reviewer, Raman is not a very sensitive method, much less sensitive than infrared. It is difficult in infrared to detect the presence of molecular H₂, so we can deduce that it will be even more difficult with Raman even if the H₂ concentration is relatively high.*
- ⇒ *Regarding H₂ however, we did not detect this species in omphacite by Raman spectroscopy, because it is probably hindered by a strong fluorescence due to Cr³⁺ beyond 3900 cm⁻¹.*

Reviewer #3 :

This manuscript has been greatly improved upon revision and it could now be published with very minor changes. In particular, the authors have carried out additional experiments which support the assignment of the 4100 cm⁻¹ band to molecular hydrogen H₂: **(1) Figure 2 shows that upon heating, the 4100 cm⁻¹ band assigned to H₂ decreases more strongly than the 4500 cm⁻¹ OH peak, as expected for the faster diffusivity of H₂; (2) Figure 2 also shows that the 4100 cm⁻¹ band intensity is proportional to the difference between total water content and water dissolved as OH.** Taken together, these observations – together with the isotopic fractionation effects observed - strongly support the presence of molecular H₂ in the samples.

There are some minor aspects that could still be improved.

The results of the heating experiments are not well presented in lines 112 to 122. The most important point here is that the 4100 cm⁻¹ H₂ band behaves differently than the 4500 cm⁻¹ OH band, confirming that these bands are caused by two different species. I would therefore reword line 120 ff: “along with a strong decrease of H₂ concentrations and a minor decrease of OH in the

Dr. Bertrand N. Moine
Lab. Magmas et Volcans
UMR6524 CNRS
Université J. Monnet
23 rue du Dr. P. Michelon
42023 Saint-Etienne Cedex 02
FRANCE

omphacite as inferred spectroscopically from the intensity of the bands at 4100 cm⁻¹ and at 2800-3600 cm⁻¹ (Figure 2).”

⇒ *We rewrote the paragraph L.123:* “This means that if any nano-water-inclusions are present, the amount of water stored in them must be negligible because otherwise, like in fluid inclusions studies, or for garnets containing water-inclusions, the speciation and concentration of bulk water would be affected^{29,33}. (2) Incremental heating of sample Obn112 up to 600°C yields an increase in the δD values of residual H along with a decrease in OH and H₂ concentrations in omphacite (Figure 4), in agreement with what is generally observed for the samples (Figure 3 A). The only way to explain this negative correlation between concentration of total water and isotopic composition is a loss during heating of a component with more negative δD values, such as H₂ or a mixture of H₂ and OH³⁴. Previous reports of such negative correlation on eclogites from Dabie Sulu have been interpreted by the loss of isotopically light molecular water due to kinetic fractionation of H-D during dehydration in the course of exhumation³⁵. Given that the ascent of kimberlites is very fast, we propose instead that the H-D isotopic fractionation is controlled by the presence of H₂. Structural H₂ is indeed observed in omphacite, and the linear relationship between the absorbance of the 4100 cm⁻¹ band and the calculated H₂ content indicates that its quantification is robust. Molecular water present in nano-inclusions seems to be negligible in these samples. However, if present in large quantities, nano-inclusions could also be filled with H₂ given that the conditions of equilibration of the present eclogites are very close to the conditions where H₂ and H₂O are miscible within the mantle³⁶.”

⇒ As shown in Figure 2, the absorbance of the 4100 cm⁻¹ band decreases with increasing temperatures and this decrease is decoupled from that of the band at ~4500 cm⁻¹, implying that these two bands cannot be attributed to the same species, i.e. to some combination of OH mode. Furthermore, the rate of the absorbance increase for the H₂ band is higher at low temperatures and becomes lower than for OH at temperatures $\geq 400^\circ\text{C}$ (see Suppl. Figure 2 & 3). This indicates that the kinetics of H₂ and H⁺ diffusion are close in this temperature range and cross over at higher temperatures in agreement with previous measurements³⁶⁻³⁹ (Figure 5). Thus, we interpret the contrasting behaviour shown in Suppl. Figure 2 & 3 as a transition from OH to H₂ during the heating stage at low temperatures following the oxidation-dehydrogenation shown in reaction 1, similar to what is inferred for natural or experimental samples^{14,40,41}.

⇒

(1)

Dr. Bertrand N. Moine
Lab. Magmas et Volcans
UMR6524 CNRS
Université J. Monnet
23 rue du Dr. P. Michelon
42023 Saint-Etienne Cedex 02
FRANCE

Such reaction is probably responsible for the stabilization of H₂ in NAMs linked to a change in iron valence in ferro-magnesian silicates, via reduction of water by ferrous iron. Ferric iron content increases in clinopyroxene and garnet with increasing pressure¹³. Thus, we can expect that eclogitic clinopyroxenes containing 7000–16000 wt. ppm Fe, with Fe³⁺/Σ Fe estimated at 20–30% due to a high jadeite component [Na⁺(Al³⁺+Fe³⁺)Si⁴⁺₂O²⁻₆]⁴², can easily incorporate the H₂ concentrations measured in this study via reaction 1. Such a reaction is common for dehydration metamorphism in subduction zones⁴⁰. However, the preservation within the mantle of high amounts of H (> 2000 ppm), considered highly mobile, at high temperatures and for a long time, needs to be explained. Diffusivity of H₂ in omphacite is currently unknown but data exist for silicate glasses, which indicate that H₂ diffusion is not very fast.

The discussion of H₂ diffusion based on the diffusion of H⁺ in line 130 is unconvincing. Yes, H₂ is larger than H⁺, but H⁺ is a charged species, such that the diffusion of H⁺ requires a counter diffusion of some other charged species to maintain local charge balance. In contrast to this, H₂ can diffuse alone with little interaction with the matrix. The mobility of H₂ should be discussed on the basis of H₂ diffusion data; while there are no such data for omphacite, there are data for silica glass, which may at least give some reasonable idea about the expected order of magnitude of the H₂ diffusion coefficient: Lee RW (1963): Diffusion of Hydrogen in Natural and Synthetic Fused Quartz. J. Chem. Phys. 38: 448.

⇒ We rephrased our previous discussion **L.162**: “For example, the diffusivity of H₂ in silica glass^{36,37} is 9.3x10⁻¹⁶ m².s⁻¹ at 23°C and 2.4x10⁻¹² m².s⁻¹ at 250°C. Using the activation energy of 44 kJ.mol⁻¹ provided in these studies we calculate a diffusivity of H₂ of 1.5x10⁻¹⁰ m².s⁻¹ at 600°C or 1.0x10⁻⁹ m².s⁻¹ at 1000°C (Figure 5). Such values are very similar to those determined for the rate of reaction (1)⁴¹ as well as for OH diffusivity in olivine³⁴ or diopside³⁵. It follows that at 20°C H₂ diffuses over 1 mm in 1.075 Ga, thus for a xenolith of 10 cm radius, a complete dehydrogenation would require 10⁴ Ga. The age of the kimberlitic eruption is 160 Ma, so we should not expect a significant H₂ degassing of the xenolith stored at near-surface conditions..”

⇒

In Figure 3 A, horizontal axis, “H₂O content” should be “total water content” for clarity.

⇒ *done*.

In Table 1 and also in Supplementary Table 2, it is important to clarify that ppm is actually “ppm by weight” and ppm H₂ refers to “ppm by weight of H₂”, not to ppm by weight of the equivalent amount of H₂O (as in many other studies).

⇒ *done*

At least in my pdf file, Supplementary Table S1 (Major element compositions) appears to be

**Dr. Bertrand N. Moine
Lab. Magmas et Volcans
UMR6524 CNRS
Université J. Monnet
23 rue du Dr. P. Michelon
42023 Saint-Etienne Cedex 02
FRANCE**

missing; there is only the caption, not the table. But Table 1 from the main article has been copied in again at the end of the supplementary material.

The English of the manuscript is generally understandable, but some minor corrections by a native English speaker would be desirable.

The manuscript has been reviewed by English native speaker (Pr. C. Harris from Cape Town University)

Bayreuth, January 22, 2020
Hans Keppler

REVIEWERS' COMMENTS:

Reviewer #1 (Remarks to the Author):

In this revised version, the authors have made corrections to the text and also provided a rebuttal to the comments. Unfortunately, there are serious mistakes with what they have done, and the critical problems casted by the previous comments are not convincingly addressed.

One of the main issues with the rebuttal and revisions of the authors is the diffusion of H₂. The very slow diffusion as argued by the authors are apparently wrong. For a grain of 1 mm size, the effective diffusion distance can be considered as 0.5 mm, then the time required for the diffusion equilibrium at different temperatures is (according to Shang et al. (2009) as used by the authors):

T(oC)	lnD	D (m ² /s)	Time (year)
20	-34.774	7.90E-16	10.03045734
100	-30.848	4.00567E-14	0.197905521
200	-27.809	8.37091E-13	0.009470232
500	-23.409	6.81953E-11	0.000116246

The time is ~10 years for the effective diffusion over 0.5 mm, that is why Hirschmann et al. (2012) stated ~40 years for the diffusion distance over 1 mm at room temperature. Apparently, the authors have made problems with their calculations (and thus their argued 1 Ga or 10000 Ga and many other arguments). Consequently, many of the relevant statements offered by the authors are wrong in both the main text and the rebuttal.

The fact that the authors did not observe any apparent loss with the intensity of their band at ~4062 cm⁻¹ at low temperature (e.g., <200 oC) would then suggest that this band may not be produced by molecular H₂. This strongly affects the discussions and conclusion, concerning both the high water contents and dD ratios. The authors should also pay very much attention to the techniques: molecular hydrogen would lose if they heat their samples prior to FTIR, SIMS or TECA analyses. Moreover, they should show convincingly why the very high contents of 'molecular hydrogen' (if it were H₂) in the samples were preserved despite the very long history of sample exhumation and the very fast diffusivity of H₂.

I would suggest the authors consider seriously the previous comments as well as the comments here, and then carry out a careful revision by addressing all the problems. They must provide convincing evidence for what has already been questioned: (1) what they have been in the spectra is indeed due to molecular hydrogen, and (2) the quantification (the different water contents between SIMS/TCEA and FTIR attributed to molecular hydrogen) is correct.

Another two issues the authors may not have realized:

1. Whatever the size of the xenolith samples (10 cm or greater), the effective diffusion distance is determined by sizes of the mineral grains. Once H₂ diffuses out of the grains (that is very fast), it would lose even fast (if not immediately) along grain boundaries.
2. I would not consider it very good if the samples were glued on glass for SIMS analyses – the H released from glue would affect the measurements and overestimate the contents. In recent years, indium has been more favored for SIMS analyses of H in NAMs and other H-bearing materials.

Xiaozhi Yang

Reviewer #2 (Remarks to the Author):

After looking at the revisions and reply letter, I see that the authors addressed the concerns of the reviewers. I am not fully convinced of their IR peak assignments until further independent proof. I also agree with the comments of reviewer 1, e.g. regarding the H₂ diffusion in minerals that could be a problem (but this is unknown... so a new area of research may appear with IR observation of H₂ in minerals?). However, their interpretation is coherent with their dataset. So I think it may be best that the ideas presented in the manuscript are assessed by a broader community and would advise publication.

However, I noticed many inconsistencies and mistakes that should be corrected or other problems that should be assessed prior to publication. See below for some of them, but the authors should read carefully their manuscript one more time to be sure to correct all inconsistencies and typos.

First, a few general comments in reply to the authors replies:

- H₂ is IR inactive by definition as this molecules presents no dipole moment, so no, IR has not been the best method to quantify H₂ to date. For H₂O/OH, it's another business that should not be mixed with H₂. As you say yourself, you can only see H₂ when it is distorted in some mineral environments, as it becomes IR active in such situations. So I trust you understand why one could be fairly skeptical when you claim you observe a molecule with IR that is, by definition, IR inactive.

- you used the fundamental O-H stretching mode in the Bolfan et al. 2014 publication to quantify OH in olivine, not the lattice mode that is a different vibrational mode.

- OK for the Raman, but one thing bothers me: why can you measure down to 75 ppm OH in olivine and not hundreds of ppm H₂ in omphacite by Raman? H₂ is visible at the hundreds of ppm level in Raman spectra of glasses... Is that because of lower H₂ Raman activity compared to the O-H stretching? Is that documented? The thing is, Raman is usually easier to perform, so if you could replicated the Bolfan study for H₂ in omphacite, this could be even better than the IR results presented here.

Some comments:

- Figure 2: why the spectrum at 600 °C seems to present more water/H₂ than that at 500 °C? Even the H₂O band is more intense... This makes no sense.

- Why the values of H₂ provided in table 1 are not equal to $H_2O_{TC} - H_2O_{tot_IR} - H_2O_{mol_IR}$? I mean, if I follow you reasoning, I do for the first line $(2750-1177-178)/18.01*2.015 = 156$ ppm H₂ and this is not equal to thye value you report, of 175 ppm? Did I miss something?

- The value for the H₂ absorption coefficient in table 1 varies a lot... This makes the impression that they are unrealistic. Maybe it is best not to report them here, and only provide a single mean value in the text with an error bar.

- Again, regarding H₂ absorption coefficients: in table 1 you have values between 0.04 and 0.10, but in the text line 177 you report a value of 0.13. Where this value comes from?

- Figure 5: very large extrapolation of element diffusive behaviour in minerals can be quite dangerous. I understand why the authors present this figure but it could be judged as not very serious by some. The authors may want to think how they could improve this figure to highlight that they are doing large extrapolations...

- there is no mention of the redox conditions of the environment. It would be good to see a comment or two to corroborate that the redox conditions are coherent with H being H₂. Could that actually be used to infer the redox condition in the mantle?

Minor comments:

- Figure 3 and 4: your y labels are not coherent, please correct the missing parenthesis.
- Figure 5: $\log D$ ($\text{m}^2 \text{s}^{-1}$) > no dot between m^2 and s^{-1} , superscript s^{-1} . X axis label unit cannot be $^\circ\text{K}$. It is Kelvin or Degree C, but no Degree K, does not exist. Please correct your legend that goes through the curves....
- the above comment for the separation between units is valid all along the manuscript... l mol^{-1} and not l.mol^{-1} ... Please correct everywhere.
- another thing, please separate all your values from their units with a space. You say 1 Kelvin so you should write 1 K. Please see the NIST handbook of chemistry for guidelines.

Reviewer #3 (Remarks to the Author):

This manuscript has now been very much improved and should be published essentially as it is. All of the referee's comments have been sufficiently addressed. The English is now also well polished.

One very small thing: In Figure 1, maybe add a note in the caption explaining from which location the two samples are coming from.

Hans Keppler

REPLY TO THE REVIEWERS' COMMENTS:

We would like to recall that the reviewers were particularly critical of the measurements and representativeness of OH and H₂ concentrations from spectroscopic methods and potential contributions from fluid inclusions. We performed high-temperature additional measurements and corrected the manuscript following their criticism. The additional measurements on the heated samples provided confirmation to our interpretations. We note, however, that the reviewers have paid little or no attention to the isotope data, which we believe to be as strong an argument as the spectroscopic data.

Reviewer #1 (Remarks to the Author):

In this revised version, the authors have made corrections to the text and also provided a rebuttal to the comments. Unfortunately, there are serious mistakes with what they have done, and the critical problems casted by the previous comments are not convincingly addressed.

One of the main issues with the rebuttal and revisions of the authors is the diffusion of H₂. The very slow diffusion as argued by the authors are apparently wrong. For a grain of 1 mm size, the effective diffusion distance can be considered as 0.5 mm, then the time required for the diffusion equilibrium at different temperatures is (according to Shang et al. (2009) as used by the authors):

T(°C)	lnD	D (m ² /s)	Time (year)
20	-34.774	7.90E-16	10.03045734
100	-30.848	4.00567E-14	0.197905521
200	-27.809	8.37091E-13	0.009470232
500	-23.409	6.81953E-11	0.000116246

The time is ~10 years for the effective diffusion over 0.5 mm, that is why Hirschmann et al. (2012) stated ~40 years for the diffusion distance over 1 mm at room temperature. Apparently, the authors have made problems with their calculations (and thus their argued 1 Ga or 10000 Ga and many other arguments). Consequently, many of the relevant statements offered by the authors are wrong in both the main text and the rebuttal.

Ok, We find the mistake in our calculation (our results stayed in seconds, not in years). We added to the text: “The validity of these estimates ultimately hinges on the knowledge of diffusion mechanisms for molecular H₂ in minerals (vacancy vs. interstitial diffusion, polaron or Franck-Turbull mechanisms), which are currently unknown. Considering that H₂ diffusivity experiments in silicate glass have shown a strong dependence (3 orders of magnitude)⁴⁶ (Figure 5) on H₂ partial pressure, the above diffusivity (hence H loss) estimates are likely to be exaggerated.”

Moreover, the diffusion model used is extremely simplified:

- 1) The diffusion coefficient is considered to depend only on temperature, in the absence of more constraints. However, the diffusion coefficient depends not only on temperature but also on the H chemical gradient and f_{O_2} - f_{H_2} . This is particularly illustrated by the data of Gaillard et al (2003) that show a high variability of the diffusion coefficient as a function of the partial pressure (fugacity) of H₂ in their experiments.
- 2) Diffusion coefficients are determined in pure H₂ atmosphere or where H₂>>H₂O. In the cratonic mantle, H speciation modelisation (Woodland & Koch 2003, Goncharov et al.2012) shows that is not the case.
- 3) The diffusion coefficient in a crystal is extremely dependent on the migration mechanism (vacancy migration, interstitial migration, self-diffusion, Franck-Turbull mechanism). It is very likely that the diffusion mechanism of molecular H₂ in a pure silica glass (interstitial migration) (Lee et al., 1963 or Shung et al.,2009), in a silicate glass (Saal et al. 2007; Gaillard et al.2013) or in a vacancy-bearing mineral (omphacite) will not be the same and will be much smaller.
- 4) It is difficult to extrapolate these diffusion values at low temperatures. However, in the absence of a change of diffusion mechanism, the knowledge of the activation energy suffices.
- 5) We don't know the influence of ionic species other than SiO₂. However, diffusion coefficients of molecular H₂ in silicate glasses may be 3 orders of magnitude lower than those determined for pure silica (Gaillard et al. 2003).
- 6) Similarly, we do not know the role of the iron oxidation reaction or its reduction involving both H⁺ and/or H₂ in the diffusion mechanism for minerals. The work on silicate glass by Saal et al., 2013 and Roskosz et al., 2018 clearly raises the issue. In view of the results on omphacite samples heated to less than 400°C, it appears that this oxidation-migration mechanism plays a role.

The diffusion of molecular H₂ in silica glass must therefore be considered as representative of the fastest kinetics and not necessarily representative of what happens in omphacite. It is very likely that diffusion coefficients in minerals are lower than those proposed for pure silica. The estimates of diffusivity (hence H loss) are therefore probably exaggerated.

The fact that the authors did not observe any apparent loss with the intensity of their band at ~4062 cm⁻¹ at low temperature (e.g., <200 °C) would then suggest that this band may not be produced by molecular H₂. This strongly affects the discussions and conclusion, concerning both the high water contents and δD ratios.

- ⇒ We lose H₂ but at higher temperatures than what you expect. We rewrote the paragraph to make things clearer about how we discard water from inclusions: “These stepwise experiments show that (1) the bulk water content and the isotopic composition of the samples are little affected by heating up to 400 °C (see **Supplementary Table 2**). Only two samples (Obn110 and Obn108) suffered up to 15% water loss during heating up to 400°C. This means that if any inclusions were present, the amount of water stored in them must be negligible or within 15% because otherwise, like in fluid inclusions studies, or for garnets containing water-inclusions, the speciation and concentration of bulk water would be affected^{29,33}.”

The authors should also pay very much attention to the techniques: molecular hydrogen would lose if they heat their samples prior to FTIR, SIMS or TECA analyses. Moreover, they should show convincingly why the very high contents of ‘molecular hydrogen’ (if it were H₂) in the samples were preserved despite the very long history of sample exhumation and the very fast diffusivity of H₂.

- ⇒ This is the first study that documents the loss of H₂ during heating of minerals.

I would suggest the authors consider seriously the previous comments as well as the comments here, and then carry out a careful revision by addressing all the problems. They must provide convincing evidence for what has already been questioned: (1) what they have seen in the spectra is indeed due to molecular hydrogen, and (2) the quantification (the different water contents between SIMS/TECA and FTIR attributed to molecular hydrogen) is correct.

- ⇒ We list below the evidences for the presence of H₂ in the samples: everything is detailed in the manuscript: We have made three major observations:

1-the difference between water contents determined by TC-EA-IRMS and FTIR point out to a missing H.

- ⇒ is it water in fluid inclusions? No because:

- SIMS measurement (which is a point beam technique) yields the same amount as TC-EA-IRMS.
- Heating of the samples evidences a small loss of water from 100 to 400°C indicating little contribution from inclusions.
- The 5200 cm⁻¹ band, related to molecular water is very weak.

- ⇒ Is it H₂? Yes because:

- 2- We observe the band at 4100 cm⁻¹ which we assign to H₂ following previous findings on silica glass or minerals
 - The intensity of this band linearly correlates with the calculated H₂ content calculated from the difference between TC-EA-IRMS and FTIR integrated in the region of OH+H₂O. The absorptivity coefficient of the 4100 cm⁻¹ is also of the same order to that determined by Shelby for glasses.
- 3- The δD increases when total H decreases: this is in agreement with isotopic fractionation between fluid and mineral, involving the difference between water contents determined by TC-EA-IRMS and FTIR point out to a missing H.

Another two issues the authors may not have realized: 1. Whatever the size of the xenolith samples (10 cm or greater), the effective diffusion distance is determined by sizes of the mineral grains. Once H₂ diffuses out of the grains (that is very fast), it would lose even fast (if not immediately) along grain boundaries.

- ⇒ You will agree that loss of H by a lonely grain floating in magma will be faster than the same grain inside a xenolith of 10-50 cm wide? Even if diffusivity at grain boundaries is faster than within grains.
- ⇒ In addition, diffusivity depends on the partial pressure of H₂, as experimentally demonstrated by Gaillard, and shown in Figure 5. At a p_{H₂} of 0.5 bars, half the atmospheric pressure, the diffusivity of H₂ in silica glass is lowered of 1 order of magnitude. What is the partial pressure of H₂ in the grain boundaries within a xenolith and “in the grain”? What is the H chemical gradient? We have no constrain.

2. I would not consider it very good if the samples were glued on glass for SIMS analyses – the H released from glue would affect the measurements and overestimate the contents. In recent years, indium has been more favored for SIMS analyses of H in NAMs and other H-bearing materials.

- ⇒ Indium is efficient to lower the background signal of H, for nominally anhydrous minerals that contain little H like garnet or olivine. Otherwise, only the background H (which is released by the environment) is measured. But for minerals that contain thousands of ppm wt H₂O like omphacite it is not so critical, we can afford an error of a few dozen of ppm.

Xiaozhi Yang

Reviewer #2 (Remarks to the Author):

After looking at the revisions and reply letter, I see that the authors addressed the concerns of the reviewers. I am not fully convinced of their IR peak assignments until further independent proof. I also agree with the comments of reviewer 1, e.g. regarding the H₂ diffusion in minerals that could be a problem (but this is unknown... so a new area of research may appear with IR observation of H₂ in minerals?). However, **their interpretation is coherent with their dataset**. So I think it may be best that the ideas presented in the manuscript are assessed by a broader community and would advice publication.

However, I noticed many inconsistencies and mistakes that should be corrected or other problems that should be assessed prior to publication. See below for some of them, but the authors should read carefully their manuscript one more time to be sure to correct all inconsistencies and typos.

First, a few general comments in reply to the authors replies:

- H₂ is IR inactive by definition as this molecules presents no dipole moment, so no, IR has not been the best method to quantify H₂ to date. For H₂O/OH, it's another business that should not be mixed with H₂. As you say yourself, you can only see H₂ when it is distorted in some mineral environments, as it becomes IR active in such situations. So I trust you understand why one could be fairly skeptical when you claim you observe a molecule with IR that is, by definition, IR inactive.

⇒ Ok we got that. We meant that H in NAMS has only been detected with IR so far.

- you used the fundamental O-H stretching mode in the Bolfan et al. 2014 publication to quantify OH in olivine, not the lattice mode that is a different vibrational mode.

⇒ Ok.

- OK for the Raman, but one thing bothers me: why can you measure down to 75 ppm OH in olivine and not hundreds of ppm H₂ in omphacite by Raman? H₂ is visible at the hundreds of ppm level in Raman spectra of glasses... Is that because of lower H₂ Raman activity compared to the O-H stretching? Is that documented? The thing is, Raman is usually easier to perform, so if you could replicated the Bolfan study for H₂ in omphacite, this could be even better than the IR results presented here.

⇒ It is not documented, but NBC often finds that OH in clinopyroxenes is not easy to detect by Raman, and they often show a huge fluorescence due to Cr.

Some comments:

- Figure 2: why the spectrum at 600 °C seems to present more water/H₂ than that at 500 °C? Even the H₂O band is more intense... This makes no sense.

⇒ We added more explanations about the behaviour of the bands with temperature: "As shown in **Figure 2**, the absorbance of the 4100 cm⁻¹ band decreases with increasing temperature. This decrease is decoupled from that of the band at ~4500 cm⁻¹ (assigned unambiguously to an Si-OH vibration) implying that these two bands cannot be attributed to the same species, i.e. the 4100 cm⁻¹ band is not the result of some combination of OH mode see **Supplementary Figures 2 & 3**). Indeed, if these two bands were due to the same species the ratio of their intensities would stay constant, which is not the case: it varies depending on temperature (see **Supplementary Figure 3**). While the integrated absorbance of the band at 4500 cm⁻¹ decreases, between 100 and 400°C, that of the 4100 cm⁻¹ band increases (**Supplementary Figure 2**). This translates into H₂ being produced while OH is being consumed. Thus, we interpret the contrasting behaviour shown in **Supplementary Figures 2 & 3** as a transition from OH to H₂ during the heating stage at low temperatures following the oxidation-dehydrogenation shown in reaction 1, similar to what is inferred for natural or experimental samples^{14,41,42}."

- Why the values of H₂ provided in table 1 are not equal to H₂O_{TC} - H₂O_{tot_IR} - H₂O_{mol_IR} ? I mean, if I follow you reasoning, I do for the first line (2750-1177-178)/18.01*2.015 = 156 ppm H₂ and this is not equal to the value you report, of 175 ppm? Did I miss something?

⇒ When we integrate for the OH concentration we also integrate for whatever molecular water too (because they vibrate at similar frequencies). The band at 5200 cm⁻¹ is informative about the amount of molecular water that the sample may contain. But if we integrate it, it would be like counting molecular water twice. We added to Table 1 caption: "Molecular H₂ was calculated by difference between total H₂O content (TC/EA-IRMS) and FTIR integrated intensity in the range of 3000-3800 cm⁻¹, accounting for both OH and potential H₂O_{molecular}."

- The value for the H₂ absorption coefficient in table 1 varies a lot... This makes the impression that they are unrealistic. Maybe it is best not to report them here, and only provide a single mean value in the text with an error bar.

⇒ We agree, it does not make much sense to calculate an extinction coefficient for all samples if some of them contain molecular H₂O. We removed the absorptivity coefficients from the Table, and explain how to calculate it using the Beer Lambert law in the Methods. We also added "If we consider only the samples annealed at the highest temperature and consider them as the most free of the contribution from inclusions then we get an

absorptivity coefficient of $30 \text{ l mol}(\text{H}_2)^{-1} \text{ cm}^{-2}$, instead of the average of $44 \text{ l mol}(\text{H}_2)^{-1} \text{ cm}^{-2}$ determined on the samples before heating at high temperature.”

- Again, regarding H₂ absorption coefficients: in table 1 you have values between 0.04 and 0.10, but in the text line 177 you report a value of 0.13. Where this value comes from?

=> We wrote “The linear absorption coefficient of H₂ that we calculate (**Table 1**) is 1500 times lower than for OH ($0.13 \text{ l mol}^{-1} \text{ cm}^{-1}$ or $\sim 44 \text{ l mol}^{-1} \text{ cm}^{-2}$ for H₂ vs. $65,000 \text{ l mol}^{-1} \text{ cm}^{-2}$ for OH³²).” The value reported in $\text{l mol}^{-1} \cdot \text{cm}^{-1}$ is the linear absorption coefficient (determined from the peak height not from the peak area) as indicated by its units. This is to be able to compare with literature data, which document also the linear absorption coefficient. While in the Table 1 it is the value measured using the peak area. We now specify “linear” absorption coefficient in the text.

- Figure 5: very large extrapolation of element diffusive behaviour in minerals can be quite dangerous. I understand why the authors present this figure but it could be judged as not very serious by some. The authors may want to think how they could improve this figure to highlight that they are doing large extrapolations...

=> we indicate in the legend of Figure 5: “dashed lines correspond to low temperature extrapolations of experimental data”

- There is no mention of the redox conditions of the environment. It would be good to see a comment or two to corroborate that the redox conditions are coherent with H being H₂. Could that actually be used to infer the redox condition in the mantle?

=> We wrote L 54: “On the other hand, the studies of eclogite xenolith from Slave, West African and Zimbabwe cratons indicate that oxygen fugacity, f_{O_2} , range to $\Delta \log f_{\text{O}_2}$ (FMQ) -2 to -4.5¹² similar to the ambient mantle^{13,14}”

Minor comments:

- Figure 3 and 4: your y labels are not coherent, please correct the missing parenthesis.

Done

- Figure 5: $\log D$ ($\text{m}^2 \text{ s}^{-1}$) > no dot between m² and s⁻¹, superscript s⁻¹. X axis label unit cannot be °K. It is Kelvin or Degree C, but no Degree K, does not exist. Please correct your legend that goes through the curves....

Done

- the above comment for the separation between units is valid all along the manuscript... l mol^{-1} and not $\text{l} \cdot \text{mol}^{-1}$... Please correct everywhere.

Done

- another thing, please separate all your values from their units with a space. You say 1 Kelvin so you should write 1 K. Please see the NIST handbook of chemistry for guidelines.

Done

Reviewer #3 (Remarks to the Author):

This manuscript has now been very much improved and should be published essentially as it is. All of the referee’s comments have been sufficiently addressed. The English is now also well polished.

One very small thing: In Figure 1, maybe add a note in the caption explaining from which location the two samples are coming from.

⇒ Done

Hans Keppler